# CcGAN: Continuous Conditional Generative Adversarial Networks for Image Generation

**Xin Ding**[*], **Yongwei Wang**[*], **Zuheng Xu, William J. Welch, Z. Jane Wang,**
The University of British Columbia
`{xin.ding@stat, yongweiw@ece, zuheng.xu@stat, will@stat,`
`zjanew@ece}.ubc.ca`

## Abstract

This work proposes the *continuous conditional generative adversarial network* (CcGAN), the first **generative** model for image generation conditional on **continuous**, **scalar** conditions (termed *regression labels*). Existing *conditional GANs* (cGANs) are mainly designed for categorical conditions (e.g., class labels); conditioning on regression labels is mathematically distinct and raises two fundamental problems: (P1) Since there may be very few (even zero) real images for some regression labels, minimizing existing empirical versions of cGAN losses (a.k.a. empirical cGAN losses) often fails in practice; (P2) Since regression labels are scalar and infinitely many, conventional label input methods (e.g., combining a hidden map of the generator/discriminator with a one-hot encoded label) are not applicable. The proposed CcGAN solves the above problems, respectively, by (S1) reformulating existing empirical cGAN losses to be appropriate for the continuous scenario; and (S2) proposing a novel method to incorporate regression labels into the generator and the discriminator. The reformulation in (S1) leads to two novel empirical discriminator losses, termed the *hard vicinal discriminator loss* (HVDL) and the *soft vicinal discriminator loss* (SVDL) respectively, and a novel empirical generator loss. The error bounds of a discriminator trained with HVDL and SVDL are derived under mild assumptions in this work. A new benchmark dataset, RC-49, is also proposed for generative image modeling conditional on regression labels. Our experiments on the Circular 2-D Gaussians, RC-49, and UTKFace datasets show that CcGAN is able to generate diverse, high-quality samples from the image distribution conditional on a given regression label. Moreover, in these experiments, CcGAN substantially outperforms cGAN both visually and quantitatively.

## 1 Introduction

*Conditional generative adversarial networks* (cGANs), first proposed in (Mirza & Osindero, 2014), aim to estimate the distribution of images conditioning on some auxiliary information, especially class labels. Subsequent studies (Odena et al., 2017; Miyato & Koyama, 2018; Brock et al., 2019; Zhang et al., 2019) confirm the feasibility of generating diverse, high-quality (even photo-realistic), and class-label consistent fake images from class-conditional GANs. Unfortunately, these cGANs do not work well for image generation with continuous, scalar conditions, termed *regression labels*, due to two problems:

**(P1)** cGANs are often trained to minimize the empirical versions of their losses (a.k.a. the empirical cGAN losses) on some training data, a principle also known as the *empirical risk minimization* (ERM) (Vapnik, 2000). The success of ERM relies on a large sample size for each distinct condition. Unfortunately, we usually have only a few real images for some regression labels. Moreover, since regression labels are continuous, some values may not even appear in the training set. Consequently, a cGAN cannot accurately estimate the image distribution conditional on such missing labels.

**(P2)** In class-conditional image generation, class labels are often encoded by one-hot vectors or label embedding and then fed into the generator and discriminator by hidden concatenation (Mirza &

---

[*]Equal contribution

Osindero, 2014), an auxiliary classifier (Odena et al., 2017) or label projection (Miyato & Koyama, 2018). A precondition for such label encoding is that the number of distinct labels (e.g., the number of classes) is finite and known. Unfortunately, in the continuous scenario, we may have infinite distinct regression labels.

A naive approach to solve **(P1)-(P2)** is to "bin" the regression labels into a series of disjoint intervals and still train a cGAN in the class-conditional manner (these interval are treated as independent classes) (Olmschenk, 2019). However, this approach has four shortcomings: (1) our experiments in Section 4 show that this approach often makes cGANs collapse; (2) we can only estimate the image distribution conditional on membership in an interval and not on the target label; (3) a large interval width leads to high label inconsistency; (4) inter-class correlation is not considered (images in successive intervals have similar distributions).

In machine learning, *vicinal risk minimization* (VRM) (Vapnik, 2000; Chapelle et al., 2001) is an alternative rule to ERM. VRM assumes that a sample point shares the same label with other samples in its vicinity. Motivated by VRM, in generative modeling conditional on regression labels where we estimate a conditional distribution $p(\boldsymbol{x}|y)$ ($\boldsymbol{x}$ is an image and $y$ is a regression label), it is natural to assume that a small perturbation to $y$ results in a negligible change to $p(\boldsymbol{x}|y)$. This assumption is consistent with our perception of the world. For example, the image distribution of facial features for a population of 15-year-old teenagers should be close to that of 16-year olds.

We therefore introduce the *continuous conditional GAN* (CcGAN) to tackle **(P1)** and **(P2)**. To our best knowledge, this is the first generative model for image generation conditional on regression labels. It is noted that Rezagholizadeh et al. (2018) and Rezagholiradeh & Haidar (2018) train GANs in an unsupervised manner and synthesize unlabeled fake images for a subsequent image regression task. Olmschenk et al. (2019) proposes a semi-supervised GAN for dense crowd counting. CcGAN is fundamentally different from these works since they do not estimate the conditional image distribution. Our contributions can be summarized as follows:

- We propose in Section 2 the CcGAN to address **(P1)** and **(P2)**, which consists of two novel empirical discriminator losses, termed the *hard vicinal discriminator loss* (HVDL) and the *soft vicinal discriminator loss* (SVDL), a novel empirical generator loss, and a novel label input method. We take the vanilla cGAN loss as an example to show how to derive HVDL, SVDL, and the novel empirical generator loss by reformulating existing empirical cGAN losses.

- We derive in Section 3 the error bounds of a discriminator trained with HVDL and SVDL.

- In Section 4, we propose a new benchmark dataset, RC-49, for the generative image modeling conditional on regression labels, since very few benchmark datasets are suitable for the studied continuous scenario. We conduct experiments on several datasets, and our experiments show that CcGAN not only generates diverse, high-quality, and label consistent images, but also substantially outperforms cGAN both visually and quantitatively.

## 2 FROM CGAN TO CCGAN

In this section, we provide the solutions **(S1)-(S2)** to **(P1)-(P2)** in a one-to-one manner by introducing the *continuous conditional GAN* (CcGAN). Please note that theoretically cGAN losses (e.g., the vanilla cGAN loss (Mirza & Osindero, 2014), the Wasserstein loss (Arjovsky et al., 2017), and the hinge loss (Miyato et al., 2018)) are suitable for both class labels and regression labels; however, their empirical versions fail in the continuous scenario (i.e., **(P1)**). Our first solution **(S1)** focuses on reformulating these empirical cGAN losses to fit into the continuous scenario. Without loss of generality, we only take the vanilla cGAN loss as an example to show such reformulation (the empirical versions of the Wasserstein loss and the hinge loss can be reformulated similarly).

The vanilla discriminator loss and generator loss (Mirza & Osindero, 2014) are defined as:

$$
\mathcal{L}(D) = -\mathbb{E}_{y \sim p_r(y)} \left[ \mathbb{E}_{\boldsymbol{x} \sim p_r(\boldsymbol{x}|y)} \left[ \log \left( D(\boldsymbol{x}, y) \right) \right] \right] - \mathbb{E}_{y \sim p_g(y)} \left[ \mathbb{E}_{\boldsymbol{x} \sim p_g(\boldsymbol{x}|y)} \left[ \log \left( 1 - D(\boldsymbol{x}, y) \right) \right] \right]
$$

$$
= - \int \log(D(\boldsymbol{x}, y)) p_r(\boldsymbol{x}, y) d\boldsymbol{x} dy - \int \log(1 - D(\boldsymbol{x}, y)) p_g(\boldsymbol{x}, y) d\boldsymbol{x} dy, \tag{1}
$$

$$\mathcal{L}(G) = -\mathbb{E}_{y \sim p_g(y)}\left[\mathbb{E}_{\boldsymbol{z} \sim q(\boldsymbol{z})}\left[\log\left(D(G(\boldsymbol{z}, y), y)\right)\right]\right] = -\int \log(D(G(\boldsymbol{z}, y), y))q(\boldsymbol{z})p_g(y)d\boldsymbol{z}dy, \tag{2}$$

where $\boldsymbol{x} \in \mathcal{X}$ is an image of size $d \times d$, $y \in \mathcal{Y}$ is a label, $p_r(y)$ and $p_g(y)$ are respectively the true and fake label marginal distributions, $p_r(\boldsymbol{x}|y)$ and $p_g(\boldsymbol{x}|y)$ are respectively the true and fake image distributions conditional on $y$, $p_r(\boldsymbol{x}, y)$ and $p_g(\boldsymbol{x}, y)$ are respectively the true and fake joint distributions of $\boldsymbol{x}$ and $y$, and $q(\boldsymbol{z})$ is the probability density function of $\mathcal{N}(\boldsymbol{0}, \boldsymbol{I})$.

Since the distributions in the losses of Eqs. (1) and (2) are unknown, for class-conditional image generation, Mirza & Osindero (2014) follows ERM and minimizes the empirical losses:

$$\widehat{\mathcal{L}}^\delta(D) = -\frac{1}{N^r}\sum_{c=1}^{C}\sum_{j=1}^{N_c^r}\log(D(\boldsymbol{x}_{c,j}^r, c)) - \frac{1}{N^g}\sum_{c=1}^{C}\sum_{j=1}^{N_c^g}\log(1 - D(\boldsymbol{x}_{c,j}^g, c)), \tag{3}$$

$$\widehat{\mathcal{L}}^\delta(G) = -\frac{1}{N^g}\sum_{c=1}^{C}\sum_{j=1}^{N_c^g}\log(D(G(\boldsymbol{z}_{c,j}, c), c)), \tag{4}$$

where $C$ is the number of classes, $N^r$ and $N^g$ are respectively the number of real and fake images, $N_c^r$ and $N_c^g$ are respectively the number of real and fake images with label $c$, $\boldsymbol{x}_{c,j}^r$ and $\boldsymbol{x}_{c,j}^g$ are respectively the $j$-th real image and the $j$-th fake image with label $c$, and the $\boldsymbol{z}_{c,j}$ are independently and identically sampled from $q(\boldsymbol{z})$. Eq. (3) implies we estimate $p_r(\boldsymbol{x}, y)$ and $p_g(\boldsymbol{x}, y)$ by their empirical probability density functions as follows:

$$\hat{p}_r^\delta(\boldsymbol{x}, y) = \frac{1}{N^r}\sum_{c=1}^{C}\sum_{j=1}^{N_c^r}\delta(\boldsymbol{x} - \boldsymbol{x}_{c,j}^r)\delta(y - c), \quad \hat{p}_g^\delta(\boldsymbol{x}, y) = \frac{1}{N^g}\sum_{c=1}^{C}\sum_{j=1}^{N_c^g}\delta(\boldsymbol{x} - \boldsymbol{x}_{c,j}^g)\delta(y - c), \tag{5}$$

where $\delta(\cdot)$ is a Dirac delta mass centered at 0. However, $\hat{p}_r^\delta(\boldsymbol{x}, y)$ and $\hat{p}_g^\delta(\boldsymbol{x}, y)$ in Eq. (5) are not good estimates in the continuous scenario because of **(P1)**.

To overcome **(P1)**, we propose a novel estimate for each of $p_r(\boldsymbol{x}, y)$ and $p_g(\boldsymbol{x}, y)$, termed the *hard vicinal estimate* (HVE). We also provide an intuitive alternative to HVE, named the *soft vicinal estimate* (SVE). The HVEs of $p_r(\boldsymbol{x}, y)$ and $p_g(\boldsymbol{x}, y)$ are:

$$\hat{p}_r^{\text{HVE}}(\boldsymbol{x}, y) = C_1 \cdot \left[\frac{1}{N^r}\sum_{j=1}^{N^r}\exp\left(-\frac{(y - y_j^r)^2}{2\sigma^2}\right)\right] \cdot \left[\frac{1}{N_{y,\kappa}^r}\sum_{i=1}^{N^r}\mathbb{1}_{\{|y-y_i^r|\leq\kappa\}}\delta(\boldsymbol{x} - \boldsymbol{x}_i^r)\right],$$

$$\hat{p}_g^{\text{HVE}}(\boldsymbol{x}, y) = C_2 \cdot \left[\frac{1}{N^g}\sum_{j=1}^{N^g}\exp\left(-\frac{(y - y_j^g)^2}{2\sigma^2}\right)\right] \cdot \left[\frac{1}{N_{y,\kappa}^g}\sum_{i=1}^{N^g}\mathbb{1}_{\{|y-y_i^g|\leq\kappa\}}\delta(\boldsymbol{x} - \boldsymbol{x}_i^g)\right], \tag{6}$$

where $\boldsymbol{x}_i^r$ and $\boldsymbol{x}_i^g$ are respectively real image $i$ and fake image $i$, $y_i^r$ and $y_i^g$ are respectively the labels of $\boldsymbol{x}_i^r$ and $\boldsymbol{x}_i^g$, $\kappa$ and $\sigma$ are two positive hyper-parameters, $C_1$ and $C_2$ are two constants making these two estimates valid probability density functions, $N_{y,\kappa}^r$ is the number of the $y_i^r$ satisfying $|y - y_i^r| \leq \kappa$, $N_{y,\kappa}^g$ is the number of the $y_i^g$ satisfying $|y - y_i^g| \leq \kappa$, and $\mathbb{1}$ is an indicator function with support in the subscript. The terms in the first square brackets of $\hat{p}_r^{\text{HVE}}$ and $\hat{p}_g^{\text{HVE}}$ imply we estimate the marginal label distributions $p_r(y)$ and $p_g(y)$ by *kernel density estimates* (KDEs) (Silverman, 1986). The terms in the second square brackets are designed based on the assumption that a small perturbation to $y$ results in negligible changes to $p_r(\boldsymbol{x}|y)$ and $p_g(\boldsymbol{x}|y)$. If this assumption holds, we can use images with labels in a small vicinity of $y$ to estimate $p_r(\boldsymbol{x}|y)$ and $p_g(\boldsymbol{x}|y)$. The SVEs of $p_r(\boldsymbol{x}, y)$ and $p_g(\boldsymbol{x}, y)$ are:

$$\hat{p}_r^{\text{SVE}}(\boldsymbol{x}, y) = C_3 \cdot \left[\frac{1}{N^r}\sum_{j=1}^{N^r}\exp\left(-\frac{(y - y_j^r)^2}{2\sigma^2}\right)\right] \cdot \left[\frac{\sum_{i=1}^{N^r}w^r(y_i^r, y)\delta(\boldsymbol{x} - \boldsymbol{x}_i^r)}{\sum_{i=1}^{N^r}w^r(y_i^r, y)}\right],$$

$$\hat{p}_g^{\text{SVE}}(\boldsymbol{x}, y) = C_4 \cdot \left[\frac{1}{N^g}\sum_{j=1}^{N^g}\exp\left(-\frac{(y - y_j^g)^2}{2\sigma^2}\right)\right] \cdot \left[\frac{\sum_{i=1}^{N^g}w^g(y_i^g, y)\delta(\boldsymbol{x} - \boldsymbol{x}_i^g)}{\sum_{i=1}^{N^g}w^g(y_i^g, y)}\right], \tag{7}$$

where $C_3$ and $C_4$ are two constants making these two estimates valid probability density functions,

$$w^r(y_i^r, y) = e^{-\nu(y_i^r - y)^2} \quad \text{and} \quad w^g(y_i^g, y) = e^{-\nu(y_i^g - y)^2}, \tag{8}$$

and the hyper-parameter $\nu > 0$. In Eq. (7), similar to the HVEs, we estimate $p_r(y)$ and $p_g(y)$ by KDEs. Instead of using samples in a hard vicinity, the SVEs use all respective samples to estimate $p_r(\boldsymbol{x}|y)$ and $p_g(\boldsymbol{x}|y)$ but each sample is assigned with a weight based on the distance of its label from $y$. Two diagrams in Fig. 1 visualize the process of using hard/soft vicinal samples to estimate $p(\boldsymbol{x}|y)$, i.e., a univariate Gaussian distribution conditional on its mean $y$.

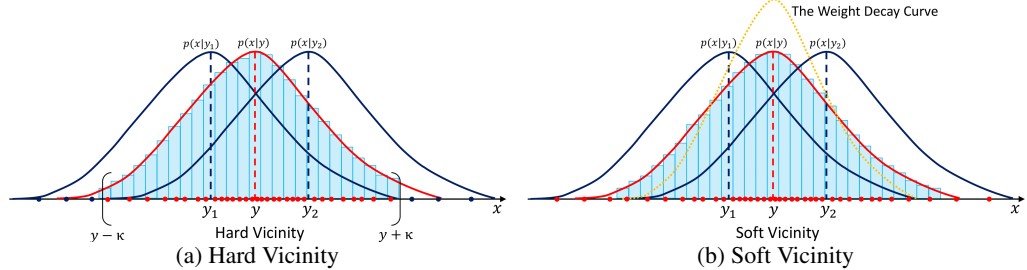

(a) Hard Vicinity        (b) Soft Vicinity

Figure 1: HVE (Eq. (6)) and SVE (Eq. (7)) estimate $p(\boldsymbol{x}|y)$ (a univariate Gaussian conditional on $y$) using two samples in hard and soft vicinities, respectively, of $y$. To estimate $p(\boldsymbol{x}|y)$ (the red Gaussian curve) only from samples drawn from $p(\boldsymbol{x}|y_1)$ and $p(\boldsymbol{x}|y_2)$ (the blue Gaussian curves), estimation is based on the samples (red dots) in a hard vicinity (defined by $y \pm \kappa$) or a soft vicinity (defined by the weight decay curve) around $y$. The histograms in blue are samples in the hard or soft vicinity. The labels $y_1$, $y$, and $y_2$ on the $x$-axis denote the means of $\boldsymbol{x}$ conditional on $y_1$, $y$, and $y_2$, respectively.

By plugging Eq. (6) and (7) into Eq. (1), we derive the *hard vicinal discriminator loss* (HVDL) and the *soft vicinal discriminator loss* (SVDL) as follows:

$$
\begin{aligned}
\widehat{\mathcal{L}}^{\text{HVDL}}(D) = &-\frac{C_5}{N^r} \sum_{j=1}^{N^r} \sum_{i=1}^{N^r} \mathbb{E}_{\epsilon^r \sim \mathcal{N}(0,\sigma^2)} \left[ \frac{\mathbb{1}_{\{|y_j^r + \epsilon^r - y_i^r| \leq \kappa\}}}{N_{y_j^r + \epsilon^r, \kappa}^r} \log(D(\boldsymbol{x}_i^r, y_j^r + \epsilon^r)) \right] \\
&-\frac{C_6}{N^g} \sum_{j=1}^{N^g} \sum_{i=1}^{N^g} \mathbb{E}_{\epsilon^g \sim \mathcal{N}(0,\sigma^2)} \left[ \frac{\mathbb{1}_{\{|y_j^g + \epsilon^g - y_i^g| \leq \kappa\}}}{N_{y_j^g + \epsilon^g, \kappa}^g} \log(1 - D(\boldsymbol{x}_i^g, y_j^g + \epsilon^g)) \right],
\end{aligned} \tag{9}
$$

$$
\begin{aligned}
\widehat{\mathcal{L}}^{\text{SVDL}}(D) = &-\frac{C_7}{N^r} \sum_{j=1}^{N^r} \sum_{i=1}^{N^r} \mathbb{E}_{\epsilon^r \sim \mathcal{N}(0,\sigma^2)} \left[ \frac{w^r(y_i^r, y_j^r + \epsilon^r)}{\sum_{i=1}^{N^r} w^r(y_i^r, y_j^r + \epsilon^r)} \log(D(\boldsymbol{x}_i^r, y_j^r + \epsilon^r)) \right] \\
&-\frac{C_8}{N^g} \sum_{j=1}^{N^g} \sum_{i=1}^{N^g} \mathbb{E}_{\epsilon^g \sim \mathcal{N}(0,\sigma^2)} \left[ \frac{w^g(y_i^g, y_j^g + \epsilon^g)}{\sum_{i=1}^{N^g} w^g(y_i^g, y_j^g + \epsilon^g)} \log(1 - D(\boldsymbol{x}_i^g, y_j^g + \epsilon^g)) \right],
\end{aligned} \tag{10}
$$

where $\epsilon^r \triangleq y - y_j^r$, $\epsilon^g \triangleq y - y_j^g$, and $C_5$, $C_6$, $C_7$, and $C_8$ are some constants.

**Generator training:** The generator of CcGAN is trained by minimizing Eq. (11),

$$\widehat{\mathcal{L}}^\epsilon(G) = -\frac{1}{N^g} \sum_{i=1}^{N^g} \mathbb{E}_{\epsilon^g \sim \mathcal{N}(0,\sigma^2)} \log(D(G(\boldsymbol{z}_i, y_i^g + \epsilon^g), y_i^g + \epsilon^g)). \tag{11}$$

**How do HVDL, SVDL, and Eq. (11) overcome (P1)?** The solution **(S1)** includes:

**(i)** Given a label $y$ as the condition, we use images in a hard/soft vicinity of $y$ to train the discriminator instead of just using images with label $y$. It enables us to estimate $p_r(\boldsymbol{x}|y)$ when there are not enough real images with label $y$.

**(ii)** From Eqs. (9) and (10), we can see that the KDEs in Eqs. (6) and (7) are adjusted by adding Gaussian noise to the labels. Moreover, in Eq. (11), we add Gaussian noise to seen labels (assume

$y_i^g$'s are seen) to train the generator to generate images at unseen labels. This enables estimation of $p_r(\boldsymbol{x}|y')$ when $y'$ is not in the training set.

**How is (P2) solved?** We propose a novel label input method. For $G$, we add the label $y$ element-wisely to the output of its first linear layer. For $D$, an extra linear layer is trained together with $D$ to embed $y$ in a latent space. We then incorporate the embedded label into $D$ by the label projection (Miyato & Koyama, 2018). Please refer to Supp. S.3 for more details.

**Remark 1.** An algorithm is proposed in Supp. S.2 for training CcGAN in practice. Moreover, CcGAN does not require any specific network architecture, therefore it can also use the state-of-art architectures in practice such as SNGAN (Miyato et al., 2018) and BigGAN (Brock et al., 2019).

## 3 ERROR BOUNDS

In this section, we derive the error bounds of a discriminator trained with $\widehat{\mathcal{L}}^{\text{HVDL}}$ and $\widehat{\mathcal{L}}^{\text{SVDL}}$ under the theoretical loss $\mathcal{L}$. First, without loss of generality, we assume $y \in [0, 1]$. Then, we introduce some notations. Let $\mathcal{D}$ stand for the *Hypothesis Space* of $D$. Let $\hat{p}_r^{\text{KDE}}(y)$ and $\hat{p}_g^{\text{KDE}}(y)$ stand for the KDEs of $p_r(y)$ and $p_g(y)$ respectively. Let $p_w^r(y'|y) \triangleq \frac{w^r(y',y)p^r(y')}{W^r(y)}$, $p_w^g(y'|y) \triangleq \frac{w^g(y',y)p^g(y')}{W^g(y)}$, $W^r(y) \triangleq \int w^r(y', y)p_r(y')dy'$ and $W^g(y) \triangleq \int w^g(y', y)p_g(y')dy'$. Denote by $D^*$ the optimal discriminator (Goodfellow et al., 2014) which minimizes $\mathcal{L}$ but may not be in $\mathcal{D}$. Let $\widetilde{D} \triangleq \arg\min_{D \in \mathcal{D}} \mathcal{L}(D)$. Let $\widehat{D}^{\text{HVDL}} \triangleq \arg\min_{D \in \mathcal{D}} \widehat{\mathcal{L}}^{\text{HVDL}}(D)$; similarly, we define $\widehat{D}^{\text{SVDL}}$.

**Definition 1.** (Hölder Class) Define the Hölder class of functions

$$\Sigma(L) \triangleq \left\{ p : \forall t_1, t_2 \in \mathcal{Y}, \exists L > 0, s.t. |p'(t_1) - p'(t_2)| \leq L|t_1 - t_2| \right\}. \tag{12}$$

Please see Supp. S.5.1 for more details of these notations. Moreover, we will also work with the following assumptions: **(A1)** All $D$'s in $\mathcal{D}$ are measurable and uniformly bounded by $U$. Let $U \triangleq \max\{\sup_{D \in \mathcal{D}} [-\log D], \sup_{D \in \mathcal{D}} [-\log(1 - D)]\}$ and $U < \infty$; **(A2)** For $\forall \boldsymbol{x} \in \mathcal{X}$ and $y, y' \in \mathcal{Y}$, $\exists g^r(\boldsymbol{x}) > 0$ and $M^r > 0$, s.t. $|p_r(\boldsymbol{x}|y') - p_r(\boldsymbol{x}|y)| \leq g^r(\boldsymbol{x})|y' - y|$ with $\int g^r(\boldsymbol{x})d\boldsymbol{x} = M^r$; **(A3)** For $\forall \boldsymbol{x} \in \mathcal{X}$ and $y, y' \in \mathcal{Y}$, $\exists g^g(\boldsymbol{x}) > 0$ and $M^g > 0$, s.t. $|p_g(\boldsymbol{x}|y') - p_g(\boldsymbol{x}|y)| \leq g^g(\boldsymbol{x})|y' - y|$ with $\int g^g(\boldsymbol{x})d\boldsymbol{x} = M^g$; **(A4)** $p_r(y) \in \Sigma(L^r)$ and $p_g(y) \in \Sigma(L^g)$.

**Theorem 1.** *Assume that (A1)-(A4) hold, then $\forall \delta \in (0, 1)$, with probability at least $1 - \delta$,*

$$\mathcal{L}(\widehat{D}^{HVDL}) - \mathcal{L}(D^*)$$

$$\leq 2U \left( \sqrt{\frac{C_{1,\delta}^{KDE} \log N^r}{N^r \sigma}} + L^r \sigma^2 \right) + 2U \left( \sqrt{\frac{C_{2,\delta}^{KDE} \log N^g}{N^g \sigma}} + L^g \sigma^2 \right) + \kappa U(M^r + M^g)$$

$$+ 2U \sqrt{\frac{1}{2} \log\left(\frac{8}{\delta}\right)} \left( \mathbb{E}_{y \sim \hat{p}_r^{KDE}(y)} \left[ \sqrt{\frac{1}{N_{y,\kappa}^r}} \right] + \mathbb{E}_{y \sim \hat{p}_g^{KDE}(y)} \left[ \sqrt{\frac{1}{N_{y,\kappa}^g}} \right] \right) + \mathcal{L}(\widetilde{D}) - \mathcal{L}(D^*),$$

$$\tag{13}$$

*for some constants $C_{1,\delta}^{KDE}$, $C_{2,\delta}^{KDE}$ depending on $\delta$.*

**Theorem 2.** *Assume that (A1)-(A4) hold, then $\forall \delta \in (0, 1)$, with probability at least $1 - \delta$,*

$$\mathcal{L}(\widehat{D}^{SVDL}) - \mathcal{L}(D^*)$$

$$\leq 2U \left( \sqrt{\frac{C_{1,\delta}^{KDE} \log N^r}{N^r \sigma}} + L^r \sigma^2 \right) + 2U \left( \sqrt{\frac{C_{2,\delta}^{KDE} \log N^g}{N^g \sigma}} + L^g \sigma^2 \right)$$

$$+ 2U \sqrt{\frac{1}{2} \log\left(\frac{16}{\delta}\right)} \left( \frac{1}{\sqrt{N^r}} \mathbb{E}_{y \sim \hat{p}_r^{KDE}(y)} \left[ \frac{1}{W^r(y)} \right] + \frac{1}{\sqrt{N^g}} \mathbb{E}_{y \sim \hat{p}_g^{KDE}(y)} \left[ \frac{1}{W^g(y)} \right] \right)$$

$$+ U \left( M^r \mathbb{E}_{y \sim \hat{p}_r^{KDE}(y)} \left[ \mathbb{E}_{y' \sim \hat{p}_w^r(y'|y)} |y' - y| \right] + M^g \mathbb{E}_{y \sim \hat{p}_g^{KDE}(y)} \left[ \mathbb{E}_{y' \sim \hat{p}_w^g(y'|y)} |y' - y| \right] \right)$$

$$+ \mathcal{L}(\widetilde{D}) - \mathcal{L}(D^*),$$

$$\tag{14}$$

*for some constant $C_{1,\delta}^{KDE}$, $C_{2,\delta}^{KDE}$ depending on $\delta$.*

**Remark 2.** The error bounds in both theorems reflect the distance of $\widehat{D}^{\text{HVDL}}$ and $\widehat{D}^{\text{SVDL}}$ from $D^*$. Enlightened by the two upper bounds, when implementing CcGAN, we should (1) avoid letting $D$ output extreme values (close to 0 or 1) so that $U$ is kept at a moderate level; (2) avoid using a too small or a too large $\kappa$ or $\nu$ to keep the third and fourth terms moderate in Eqs. (13) and (14). Please see Supp. S.5.2.5 for a more detailed interpretation and Supp. S.5.2 for the proofs.

## 4 EXPERIMENT

In this section, we study the effectiveness of CcGAN on three datasets where cGAN (Mirza & Osindero, 2014) cannot generate realistic samples. For a fair comparison, cGAN and CcGAN use the same network architecture (a customized architecture for Circular 2-D Gaussians and the SNGAN (Miyato et al., 2018) architecture for RC-49 and UTKFace) except for the label input modules. For stability, image labels are normalized to $[0, 1]$ in the RC-49 and UTKFace datasets during training.

### 4.1 CIRCULAR 2-D GAUSSIANS

We first test on the synthetic data generated from 120 2-D Gaussians with different means.

**Experimental setup:** The means of the 120 Gaussians are evenly arranged on a unit circle centered at the origin $O$ of a 2-D space. The Gaussians share a common covariance matrix $\tilde{\sigma}^2 I_{2\times 2}$, where $\tilde{\sigma} = 0.02$. We generate 10 samples from each Gaussian for training. Fig. 2a shows 1,200 training samples (blue dots) from these Gaussians with their means (red dots) on a unit circle. The unit circle can be seen as a clock where we take the mean at 12 o'clock (point $A$) as the baseline point. Given another mean on the circle (point $B$), the label $y$ for samples generated from the Gaussian with mean $B$ is defined as the clockwise angle (in radians) between line segments $OA$ and $OB$. E.g., the label for samples from the Gaussian at $A$ is 0. Both cGAN and our proposed CcGAN are trained on this training set. When implementing cGAN, angles are treated as class labels (each Gaussian is treated as a class); while when implementing CcGAN, angles are treated as real numbers. The network architectures of cGAN and CcGAN are shown in Supp. S.6.1. Both cGAN and CcGAN are trained for 6,000 iterations. We use the rule of thumb formulae in Supp. S.4 to select the hyper-parameters of HVDL and SVDL, i.e., $\sigma \approx 0.074$, $\kappa \approx 0.017$ and $\nu = 3600$ (see Supp. S.6.2 for details).

For testing, we choose 360 points evenly distributed on the unit circle as the means of 360 Gaussians. For each Gaussian, we generate 100 samples, yielding a test set with 36,000 samples. It should be noted that, among these 360 Gaussians, at least 240 are not used at the training. In other words, there are at least 240 labels in the testing set which do not appear in the training set. For each test angle, we generate 100 fake samples from each trained GAN, yielding 36,000 fake samples from each GAN in total. The quality of these fake samples is evaluated. We repeat the whole experiment three times and report in Table 1 the average quality over three repetitions.

**Evaluation metrics and quantitative results:** In the label-conditional scenario, each fake sample $x$ with label $y$ is compared with the mean $(\sin(y), \cos(y))$ of a Gaussian on the unit circle with label $y$. A fake sample is defined as "high-quality" if its Euclidean distance from $x$ to $(\sin(y), \cos(y))$ is smaller than $4\tilde{\sigma} = 0.08$. A mode (i.e., a Gaussian) is said to be recovered if at least one high-quality sample is assigned to it. We also measure the quality of fake samples with label $y$ by computing the 2-Wasserstein Distance ($\mathcal{W}_2$) (Peyré et al., 2019) between $p_r(x|y) = \mathcal{N}([\sin(y), \cos(y)]^\intercal, \tilde{\sigma} I)$ and $p_g(x|y) = \mathcal{N}(\mu_y^g, \Sigma_y^g)$, where we assume $p_g(x|y)$ is Gaussian and its mean and covariance are estimated by the sample mean and sample covariance of 100 fake samples with label $y$. In Table 1, we report the average percentage of high-quality fake samples and the average percentage of recovered modes over 3 repetitions. We also report the average $\mathcal{W}_2$ over 360 testing angles. We can see CcGAN substantially outperforms cGAN.

**Visual results:** We select 12 angles which do not appear in the training set. We then use cGAN and CcGAN to generate 100 samples for each unobserved angle. Fig. 2 visually confirms the obervation from the numerical metrics: the fake samples from the two CcGAN methods are more realistic.

### 4.2 RC-49

Since most benchmark datasets in the GAN literature do not have continuous, scalar regression labels, we propose a new benchmark dataset—RC-49, a synthetic dataset created by rendering 49 3-D chair

Table 1: Average quality of 36,000 fake samples from cGAN and CcGAN over three repetitions with standard deviations after the "±" symbol. "↓" ("↑") indicates lower (higher) values are preferred.

| Method | % High Quality ↑ | % Recovered Modes ↑ | 2-Wasserstein Dist. ↓ |
|---|---|---|---|
| cGAN (120 classes) | $68.8 \pm 4.8$ | $81.8 \pm 3.9$ | $3.32 \times 10^{-2} \pm 3.13 \times 10^{-2}$ |
| CcGAN (HVDL) | $\mathbf{99.3 \pm 0.4}$ | $\mathbf{100.0 \pm 0.0}$ | $\mathbf{3.03 \times 10^{-4} \pm 5.05 \times 10^{-5}}$ |
| CcGAN (SVDL) | $\mathbf{99.6 \pm 0.1}$ | $\mathbf{100.0 \pm 0.0}$ | $\mathbf{2.56 \times 10^{-4} \pm 8.95 \times 10^{-6}}$ |

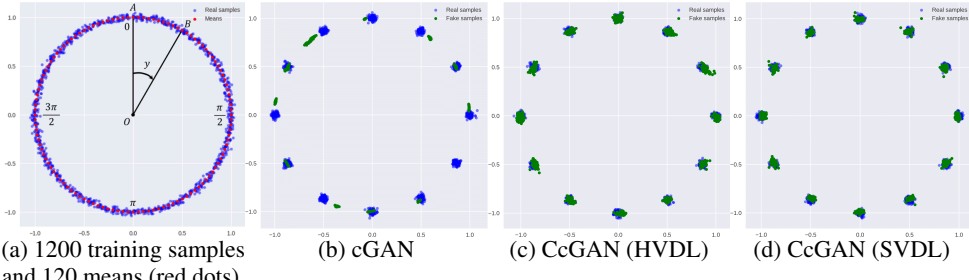

(a) 1200 training samples and 120 means (red dots).     (b) cGAN     (c) CcGAN (HVDL)     (d) CcGAN (SVDL)

Figure 2: Visual results for the Circular 2-D Gaussians simulation. (a) shows 1,200 training samples from 120 Gaussians, with 10 samples per Gaussian. In (b) to (d), each GAN generates 100 fake samples at each of 12 means not appearing in the training set, where green and blue dots stand for fake and real samples respectively.

models at different yaw angles. Each of 49 chair models is rendered at 899 yaw angles ranging from 0.1 to 89.9 with step size 0.1. Therefore, RC-49 consists of 44,051 $64 \times 64$ rendered RGB images and 899 distinct angles. Please see Supp. S.7 for more details of the data generation. Some example images are shown in Fig. 3.

**Experimental setup:** Not all images are used for the cGAN and CcGAN training. A yaw angle is selected for training if its last digit is odd. Moreover, at each selected angle, only 25 images are randomly chosen for training. Thus, the training set includes 11250 images and 450 distinct angles. The remaining images are held out for evaluation.

When training cGAN, we divide $[0.1, 89.9]$ into 150 equal intervals where each interval is treated as a class. When training CcGAN, we use the rule of thumb formulae in Supp. S.4 to select the three hyper-parameters of HVDL and SVDL, i.e., $\sigma \approx 0.047$, $\kappa \approx 0.004$ and $\nu = 50625$. Both cGAN and CcGAN are trained for 30,000 iterations with batch size 256. Afterwards, we evaluate the trained GANs on all 899 angles by generating 200 fake images for each angle. Please see Supp. S.7 for the network architectures and more details about the training/testing setup.

**Quantitative and visual results:** To evaluate (1) the visual quality, (2) the intra-label diversity, and (3) the label consistency (whether assigned labels of fake images are consistent with their true labels) of fake images, we study an overall metric and three separate metrics here. (i) **Intra-FID** (Miyato & Koyama, 2018) is utilized as the overall metric. It computes the *Fréchet inception distance* (FID) (Heusel et al., 2017) separately at each of the 899 evaluation angles and reports the average FID score. (ii) **Naturalness Image Quality Evaluator (NIQE)** (Mittal et al., 2012) measures the visual quality only. (iii) **Diversity** is the average entropy of predicted chair types of fake images over evaluation angles. (iv) **Label Score** is the average absolute error between assigned labels and predicted labels. Please see Supp. S.7.5 for details of these metrics.

We report in Table 2 the performances of each GAN. The example fake images in Fig. 3 and line graphs in Fig. 5 support the quantitative results. cGAN often generates unrealistic, identical images for a target angle (i.e., low visual quality and low intra-label diversity). "Binning" $[0.1, 89.9]$ into other number of classes (e.g., 90 classes and 210 classes) is also tried but does not improve cGAN's performance. In contrast, strikingly better visual quality and higher intra-label diversity of both CcGAN methods are visually evident. Please note that CcGAN is designed to sacrifice some (not too much) label consistency for better visual quality and higher diversity, and this explains why CcGAN does not outperform cGAN in terms of the label score in Table 2.

Table 2: Average quality of 179,800 fake RC-49 images from cGAN and CcGAN with standard deviations after the "±" symbol. "↓" ("↑") indicates lower (higher) values are preferred.

| Method | Intra-FID ↓ | NIQE ↓ | Diversity ↑ | Label Score ↓ |
|---|---|---|---|---|
| cGAN (150 classes) | $1.720 \pm 0.384$ | $2.731 \pm 0.162$ | $0.779 \pm 0.199$ | $\mathbf{4.815 \pm 5.152}$ |
| CcGAN (HVDL) | $\mathbf{0.612 \pm 0.145}$ | $\mathbf{1.869 \pm 0.181}$ | $\mathbf{2.353 \pm 0.121}$ | $5.617 \pm 4.452$ |
| CcGAN (SVDL) | $\mathbf{0.515 \pm 0.181}$ | $\mathbf{1.853 \pm 0.159}$ | $\mathbf{2.610 \pm 0.113}$ | $4.982 \pm 4.439$ |

### 4.3 UTKFACE

In this section, we compare CcGAN and cGAN on UTKFace (Zhang et al., 2017), a dataset consisting of RGB images of human faces which are labeled by age.

**Experimental setup:** In this experiment, we only use images with age in $[1, 60]$. Some images with bad visual quality and watermarks are also discarded. After the preprocessing, 14,760 images are left. The number of images for each age ranges from 50 to 1051. We resize all selected images to $64 \times 64$. Some example UTKFace images are shown in the first image array in Fig.4.

When implementing cGAN, each age is treated as a class. For CcGAN we use the rule of thumb formulae in Supp. S.4 to select the three hyper-parameters of HVDL and SVDL, i.e., $\sigma \approx 0.041$, $\kappa \approx 0.017$ and $\nu = 3600$. Both cGAN and CcGAN are trained for 40,000 iterations with batch size 512. In testing, we generate 1,000 fake images from each trained GAN for each age. Please see Supp. S.8 for more details of data preprocessing, network architectures and training/testing setup.

**Quantitative and visual results:** Similar to the RC-49 experiment, we evaluate the quality of fake images by Intra-FID, NIQE, Diversity (entropy of predicted races), and Label Score. We report in Table 3 the average quality of 60,000 fake images. We also show in Fig. 4 some example fake images from cGAN and CcGAN and line graphs of FID/NIQE versus ages in Fig. 5. Analogous to the quantitative comparisons, we can see that CcGAN performs much better than cGAN.

Table 3: Average quality of 60,000 fake UTKFace images from cGAN and CcGAN with standard deviations after the "±" symbol. "↓" ("↑") indicates lower (higher) values are preferred.

| Method | Intra-FID ↓ | NIQE ↓ | Diversity ↑ | Label Score ↓ |
|---|---|---|---|---|
| cGAN (60 classes) | $4.516 \pm 0.965$ | $2.315 \pm 0.306$ | $0.254 \pm 0.353$ | $11.087 \pm 8.119$ |
| CcGAN (HVDL) | $\mathbf{0.572 \pm 0.167}$ | $\mathbf{1.739 \pm 0.145}$ | $\mathbf{1.338 \pm 0.178}$ | $\mathbf{9.782 \pm 7.166}$ |
| CcGAN (SVDL) | $\mathbf{0.547 \pm 0.181}$ | $\mathbf{1.753 \pm 0.196}$ | $\mathbf{1.326 \pm 0.198}$ | $10.739 \pm 8.340$ |

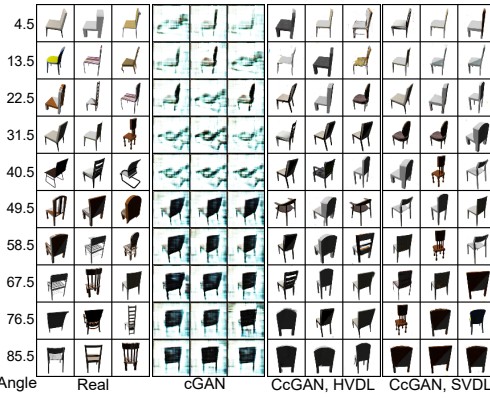 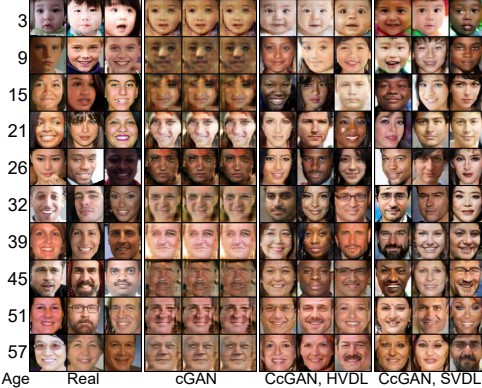

Figure 3: Three RC-49 example images for each of 10 angles: real images and example fake images from cGAN and two proposed CcGANs, respectively. CcGANs produce chair images with **higher visual quality and more diversity**.

Figure 4: Three UTKFace example images for each of 10 ages: real images and example fake images from cGAN and two proposed CcGANs, respectively. CcGANs produce face images with **higher visual quality and more diversity**.

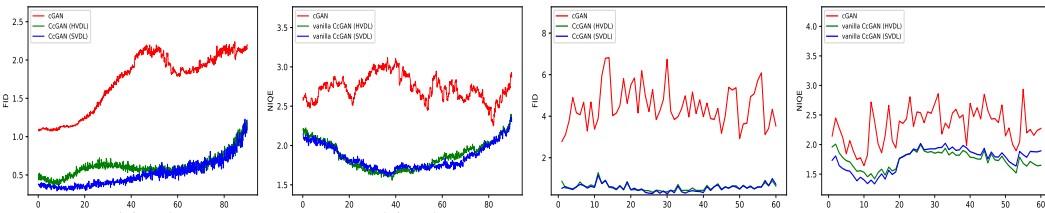

(a) RC-49: FID vs Angle  (b) RC-49: NIQE vs Angle  (c) UTKFace: FID vs Age  (d) UTKFace: NIQE vs Age

Figure 5: Line graphs of FID/NIQE versus regression labels on RC-49 and UTKFace. Figs. 5(a) to 5(d) show that two CcGANs consistently outperform cGAN across all regression labels. The graphs of CcGANs also appear smoother than those of cGAN because of HVDL and SVDL.

## 5 CONCLUSION

As the first generative model, we propose the CcGAN in this paper for image generation conditional on regression labels. In CcGAN, two novel empirical discriminator losses (HVDL and SVDL), a novel empirical generator loss and a novel label input method are proposed to overcome the two problems of existing cGANs. The error bounds of a discriminator trained under HVDL and SVDL are studied in this work. A new benchmark dataset, RC-49, is also proposed for the continuous scenario. Finally we demonstrate the superiority of the proposed CcGAN to cGAN on the Circular 2-D Gaussians, RC-49, and UTKFace datasets.

### ACKNOWLEDGMENTS

This work was supported by the Natural Sciences and Engineering Research Council of Canada (NSERC) under Grants CRDPJ 476594-14, RGPIN-2019-05019, and RGPAS2017-507965.

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

SUPPLEMENTARY MATERIAL

## S.1  GITHUB REPOSITORY

Please find the codes for this paper at Github:

https://github.com/UBCDingXin/improved_CcGAN

## S.2  ALGORITHMS FOR CCGAN TRAINING

---

**Algorithm 1:** An algorithm for CcGAN training with the proposed HVDL.

---

**Data:** $N^r$ real image-label pairs $\Omega^r = \{(\boldsymbol{x}_1^r, y_1^r), \ldots, (\boldsymbol{x}_{N^r}^r, y_{N^r}^r)\}$, $N_{uy}^r$ ordered distinct labels
$\Upsilon = \{y_{[1]}^r, \ldots, y_{[N_{uy}^r]}^r\}$ in the dataset, preset $\sigma$ and $\kappa$, number of iterations $K$, the discriminator
batch size $m^d$, and the generator batch size $m^g$.

**Result:** Trained generator $G$.

1 **for** $k = 1$ **to** $K$ **do**
2     **Train D**;
3     Draw $m^d$ labels $Y^d$ with replacement from $\Upsilon$;
4     Create a set of target labels $Y^{d,\epsilon} = \{y_i + \epsilon | y_i \in Y^d, \epsilon \in \mathcal{N}(0, \sigma^2), i = 1, \ldots, m^d\}$ ($D$ training is
     conditional on these labels) ;
5     Initialize $\Omega_d^r = \phi, \Omega_d^f = \phi$;
6     **for** $i = 1$ **to** $m^d$ **do**
7         Randomly choose an image-label pair $(\boldsymbol{x}, y) \in \Omega^r$ satisfying $|y - y_i - \epsilon| \leq \kappa$ where
         $y_i + \epsilon \in Y^{d,\epsilon}$ and let $\Omega_d^r = \Omega_d^r \cup (\boldsymbol{x}, y_i + \epsilon)$. ;
8         Randomly draw a label $y'$ from $U(y_i + \epsilon - \kappa, y_i + \epsilon + \kappa)$ and generate a fake image $\boldsymbol{x}'$ by
         evaluating $G(\boldsymbol{z}, y')$, where $\boldsymbol{z} \sim \mathcal{N}(\boldsymbol{0}, \boldsymbol{I})$. Let $\Omega_d^f = \Omega_d^f \cup (\boldsymbol{x}', y_i + \epsilon)$. ;
9     **end**
10     Update $D$ with samples in set $\Omega_d^r$ and $\Omega_d^f$ via gradient-based optimizers based on Eq.(6);
11     **Train G**;
12     Draw $m^g$ labels $Y^g$ with replacement from $\Upsilon$;
13     Create another set of target labels $Y^{g,\epsilon} = \{y_i + \epsilon | y_i \in Y^g, \epsilon \in \mathcal{N}(0, \sigma^2), i = 1, \ldots, m^g\}$ ($G$
     training is conditional on these labels) ;
14     Generate $m^g$ fake images conditional on $Y^{g,\epsilon}$ and put these image-label pairs in $\Omega_g^f$ ;
15     Update $G$ with samples in $\Omega_g^f$ via gradient-based optimizers based on Eq.(11) ;
16 **end**

---

**Remark S.3.** It should be noted that, for computational efficiency, the normalizing constants $N_{y_j^r + \epsilon^r, \kappa}^r$, $N_{y_j^g + \epsilon^g, \kappa}^g$, $\sum_{i=1}^{N^r} w^r(y_i^r, y_j^r + \epsilon^r)$, and $\sum_{i=1}^{N^g} w^g(y_i^g, y_j^g + \epsilon^g)$ in Eq. (9) and (10) are excluded from the training and only used for theoretical analysis.

## S.3  MORE DETAILS OF THE PROPOSED LABEL INPUT METHOD IN SECTION 2

We propose a novel way to input labels to the conditional generative adversarial networks. For the generator, we add a regression label element-wise to the feature map of the first linear layer. For the discriminator, labels are first projected to a latent space learned by an extra linear layer. Then, we incorporate the embedded labels into the discriminator by the label projection (Miyato & Koyama, 2018). Figs. S.3.6 and S.3.7 visualizes our proposed label input method. Please refer to our codes for more details.

## S.4  A RULE OF THUMB FOR HYPER-PARAMETER SELECTION

In our experiments, we normalize labels to real numbers in $[0, 1]$ and the hyper-parameter selection is conducted based on the normalized labels. To be more specific, the hyper-parameter $\sigma$ is computed based on a rule-of-thumb formula for the bandwidth selection of KDE (Silverman, 1986), i.e.,

---

**Algorithm 2:** An algorithm for CcGAN training with the proposed SVDL.

---

**Data:** $N^r$ real image-label pairs $\Omega^r = \{(\boldsymbol{x}_1^r, y_1^r), \ldots, (\boldsymbol{x}_{N^r}^r, y_{N^r}^r)\}$, $N_{uy}^r$ ordered distinct labels $\Upsilon = \{y_{[1]}^r, \ldots, y_{[N_{uy}^r]}^r\}$ in the dataset, preset $\sigma$ and $\nu$, number of iterations $K$, the discriminator batch size $m^d$, and the generator batch size $m^g$.

**Result:** Trained generator $G$.

1 **for** $k = 1$ **to** $K$ **do**
2     **Train D**;
3     Draw $m^d$ labels $Y^d$ with replacement from $\Upsilon$;
4     Create a set of target labels $Y^{d,\epsilon} = \{y_i + \epsilon | y_i \in Y^d, \epsilon \in \mathcal{N}(0, \sigma^2), i = 1, \ldots, m^d\}$ ($D$ training is conditional on these labels) ;
5     Initialize $\Omega_d^r = \phi, \Omega_d^f = \phi$;
6     **for** $i = 1$ **to** $m^d$ **do**
7         Randomly choose an image-label pair $(\boldsymbol{x}, y) \in \Omega^r$ satisfying $e^{-\nu(y - y_i - \epsilon)^2} > 10^{-3}$ where $y_i + \epsilon \in Y^{d,\epsilon}$ and let $\Omega_d^r = \Omega_d^r \cup (\boldsymbol{x}, y_i + \epsilon)$. This step is used to exclude real images with too small weights. ;
8         Compute $w_i^r(y, y_i + \epsilon) = e^{-\nu(y_i + \epsilon - y)^2}$;
9         Randomly draw a label $y'$ from $U(y_i + \epsilon - \sqrt{-\frac{\log 10^{-3}}{\nu}}, y_i + \epsilon + \sqrt{-\frac{\log 10^{-3}}{\nu}})$ and generate a fake image $\boldsymbol{x}'$ by evaluating $G(\boldsymbol{z}, y')$, where $\boldsymbol{z} \sim \mathcal{N}(\boldsymbol{0}, \boldsymbol{I})$. Let $\Omega_d^f = \Omega_d^f \cup (\boldsymbol{x}', y_i + \epsilon)$. ;
10         Compute $w_i^g(y', y_i + \epsilon) = e^{-\nu(y_i + \epsilon - y')^2}$;
11     **end**
12     Update $D$ with samples in set $\Omega_d^r$ and $\Omega_d^f$ via gradient-based optimizers based on Eq.(7);
13     **Train G**;
14     Draw $m^g$ labels $Y^g$ with replacement from $\Upsilon$;
15     Create another set of target labels $Y^{g,\epsilon} = \{y_i + \epsilon | y_i \in Y^g, \epsilon \in \mathcal{N}(0, \sigma^2), i = 1, \ldots, m^g\}$ ($G$ training is conditional on these labels) ;
16     Generate $m^g$ fake images conditional on $Y^{g,\epsilon}$ and put these image-label pairs in $\Omega_g^f$ ;
17     Update $G$ with samples in $\Omega_g^f$ via gradient-based optimizers based on Eq.(11) ;
18 **end**

---

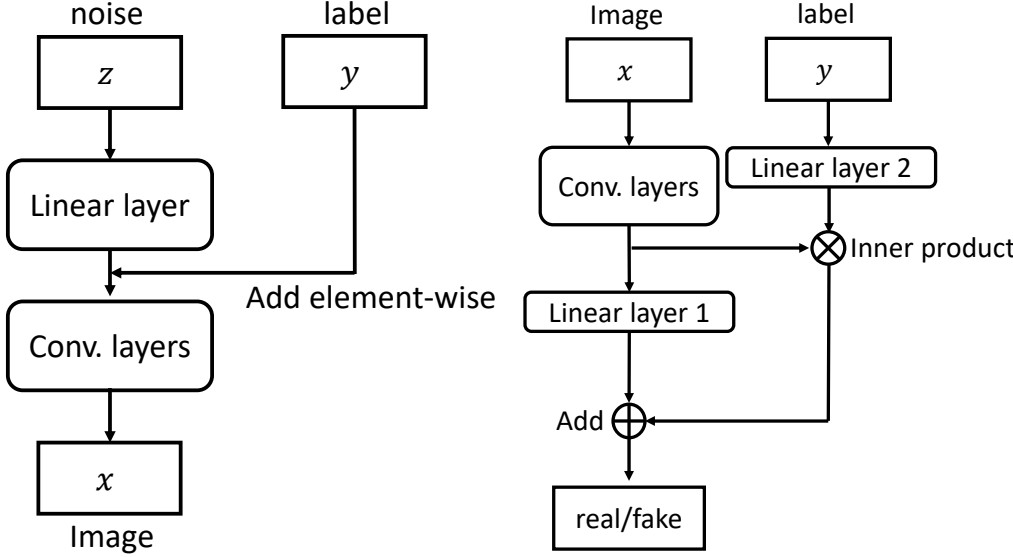

Figure S.3.6: The label input method for the generator in CcGAN.

Figure S.3.7: The label input method for the discriminator in CcGAN.

$\sigma = \left(4\hat{\sigma}_{y^r}^5 / 3N^r\right)^{1/5}$, where $\hat{\sigma}_{y^r}$ is the sample standard deviation of normalized labels in the training set. Let $\kappa_{\text{base}} = \max\left(y_{[2]}^r - y_{[1]}^r, y_{[3]}^r - y_{[2]}^r, \ldots, y_{[N_{uy}^r]}^r - y_{[N_{uy}^r - 1]}^r\right)$, where $y_{[l]}^r$ is the $l$-th smallest

normalized distinct real label and $N_{\text{uy}}^r$ is the number of normalized distinct labels in the training set. The $\kappa$ is set as a multiple of $\kappa_{\text{base}}$ (i.e., $\kappa = m_\kappa \kappa_{\text{base}}$) where the multiplier $m_\kappa$ stands for 50% of the minimum number of neighboring labels used for estimating $p_r(\boldsymbol{x}|y)$ given a label $y$. For example, $m_\kappa = 1$ implies using 2 neighboring labels (one on the left while the other one on the right). In our experiments, $m_\kappa$ is generally set as 1 or 2. In some extreme case when many distinct labels have too few real samples, we may consider increasing $m_\kappa$. We also found $\nu = 1/\kappa^2$ works well in our experiments.

## S.5 MORE DETAILS OF THEOREMS S.4 AND S.5

### S.5.1 SOME NECESSARY DEFINITIONS AND NOTATIONS

- The hypothesis space $\mathcal{D}$ is a set of functions that can be represented by $D$ (a neural network with determined architecture).

- In the HVDL case, denote by $p_r^{y,\kappa}(\boldsymbol{x}) \triangleq \int_{y-\kappa}^{y+\kappa} p_r(\boldsymbol{x}|y')p_r(y')dy'$ the marginal distribution of real images with labels in $[y-\kappa, y+\kappa]$ and similarly to $p_g^{y,\kappa}(\boldsymbol{x})$ of fake images.

- In the SVDL case, given $y$ and weight functions (E.q. (8)), if the number of real and fake images are infinite, the empirical density converges to $p_r^{y,w^r}(\boldsymbol{x}) \triangleq \int p_r(\boldsymbol{x}|y')\frac{w^r(y',y)p_r(y')}{W^r(y)}dy'$ and $p_g^{y,w^g}(\boldsymbol{x}) \triangleq \int p_g(\boldsymbol{x}|y')\frac{w^g(y',y)p_g(y)}{W^g(y)}dy'$ respectively, where $W^r(y) \triangleq \int w^r(y',y)p_r(y')dy'$ and $W^g(y) \triangleq \int w^g(y',y)p_g(y')dy'$.

- Let $p_w^r(y'|y) \triangleq \frac{w^r(y',y)p^r(y')}{W^r(y)}$ and $p_w^g(y'|y) \triangleq \frac{w^g(y',y)p^g(y')}{W^g(y)}$.

- The Hölder Class defined in Definition 1 is a set of functions with bounded second derivatives, which controls the variation of the function when parameter changes. (A4) implies the two probability density functions $p_r(y)$ and $p_g(y)$ are assumed in the Hölder Class.

- Given a $G$, the optimal discriminator which minimizes $\mathcal{L}$ is in the form of

$$D^*(\boldsymbol{x}, y) = \frac{p_r(\boldsymbol{x}, y)}{p_r(\boldsymbol{x}, y) + p_g(\boldsymbol{x}, y)}. \tag{S.15}$$

However, $D^*$ may not be covered by the hypothesis space $\mathcal{D}$. The $\widetilde{D}$ is the minimizer of $\mathcal{L}$ in the hypothesis space $\mathcal{D}$. Thus, $\mathcal{L}(\widetilde{D}) - \mathcal{L}(D^*)$ should be a non-negative constant. In CcGAN, we minimize $\widehat{\mathcal{L}}^{\text{HVDL}}(D)$ or $\widehat{\mathcal{L}}^{\text{HVDL}}(D)$ with respect to $D \in \mathcal{D}$, so we are more interested in the distance of $\widehat{D}^{\text{HVDL}}$ and $\widehat{D}^{\text{SVDL}}$ from $D^*$, i.e., $\mathcal{L}(\widehat{D}^{\text{HVDL}}) - \mathcal{L}(D^*)$ and $\mathcal{L}(\widehat{D}^{\text{SVDL}}) - \mathcal{L}(D^*)$.

### S.5.2 PROOFS OF THEOREMS 1 AND 2

#### S.5.2.1 TECHNICAL LEMMAS

Before we move to the proofs of Theorems 1 and 2, we provide several technical lemmas used in the later proof.

Recall notations and assumptions in Sections 3 and S.5.1, then we derive the following lemmas.

**Lemma S.1.** *Suppose that (A1)-(A2) and (A4) hold, then $\forall \delta \in (0,1)$, with probability at least $1 - \delta$,*

$$\sup_{D \in \mathcal{D}} \left| \frac{1}{N_{y,\kappa}^r} \sum_{i=1}^{N^r} \mathbb{1}_{\{|y-y_i^r|\leq\kappa\}} \left[-\log D(\boldsymbol{x}_i^r, y)\right] - \mathbb{E}_{\boldsymbol{x} \sim p_r(\boldsymbol{x}|y)} \left[-\log D(\boldsymbol{x}, y)\right] \right|$$

$$\leq U\sqrt{\frac{1}{2N_{y,\kappa}^r} \log\left(\frac{2}{\delta}\right)} + \frac{\kappa U M^r}{2}, \tag{S.16}$$

*for a given $y$.*

*Proof.* Triangle inequality yields

$$\sup_{D \in \mathcal{D}} \left| \frac{1}{N_{y,\kappa}^r} \sum_{i=1}^{N^r} \mathbb{1}_{\{|y - y_i^r| \leq \kappa\}} \left[ -\log D(\boldsymbol{x}_i^r, y) \right] - \mathbb{E}_{\boldsymbol{x} \sim p_r(\boldsymbol{x}|y)} \left[ -\log D(\boldsymbol{x}, y) \right] \right|$$

$$\leq \sup_{D \in \mathcal{D}} \left| \frac{1}{N_{y,\kappa}^r} \sum_{i=1}^{N^r} \mathbb{1}_{\{|y - y_i^r| \leq \kappa\}} \left[ -\log D(\boldsymbol{x}_i^r, y) \right] - \mathbb{E}_{\boldsymbol{x} \sim p_r^{y,\kappa}(\boldsymbol{x})} \left[ -\log D(\boldsymbol{x}, y) \right] \right|$$

$$+ \sup_{D \in \mathcal{D}} \left| \mathbb{E}_{\boldsymbol{x} \sim p_r^{y,\kappa}(\boldsymbol{x})} \left[ -\log D(\boldsymbol{x}, y) \right] - \mathbb{E}_{\boldsymbol{x} \sim p_r(\boldsymbol{x}|y)} \left[ -\log D(\boldsymbol{x}, y) \right] \right|$$

We then bound the two terms of the RHS separately as follows:

1. Real images with labels in $[y - \kappa, k + \kappa]$ can be seen as independent samples from $p_r^{y,\kappa}(\boldsymbol{x})$. Then the first term can be bounded by applying Hoeffding's inequality as follows: $\forall \delta \in (0, 1)$, with at least probability $1 - \delta$,

$$\sup_{D \in \mathcal{D}} \left| \frac{1}{N_{y,\kappa}^r} \sum_{i=1}^{N^r} \mathbb{1}_{\{|y - y_i^r| \leq \kappa\}} \left[ U \frac{-\log D(\boldsymbol{x}_i^r, y)}{U} \right] - \mathbb{E}_{\boldsymbol{x} \sim p_r^{y,\kappa}(\boldsymbol{x})} \left[ U \frac{-\log D(\boldsymbol{x}, y)}{U} \right] \right|$$

$$\leq U \sqrt{\frac{1}{2 N_{y,\kappa}^r} \log \left( \frac{2}{\delta} \right)}.$$

(S.17)

2. For the second term, by the definition of $p_r^{y,\kappa}(\boldsymbol{x})$ and defining $p_\kappa(y') = \frac{\mathbb{1}_{\{|y' - y| \leq \kappa\}} p(y')}{\int \mathbb{1}_{\{|y' - y| \leq \kappa\}} p(y') dy'}$, we have

$$\sup_{D \in \mathcal{D}} \left| \mathbb{E}_{\boldsymbol{x} \sim p_r^{y,\kappa}(\boldsymbol{x})} \left[ -\log D(\boldsymbol{x}, y) \right] - \mathbb{E}_{\boldsymbol{x} \sim p_r(\boldsymbol{x}|y)} \left[ -\log D(\boldsymbol{x}, y) \right] \right|$$

(by the definition of total variation and the boundness of $-\log D$)

$$\leq \frac{U}{2} \int \left| p_r^{y,\kappa}(\boldsymbol{x}) - p_r(\boldsymbol{x}|y) \right| d\boldsymbol{x}.$$

(S.18)

Then, focusing on $|p_r^{y,\kappa}(\boldsymbol{x}) - p_r(\boldsymbol{x}|y)|$,

$$|p_r^{y,\kappa}(\boldsymbol{x}) - p_r(\boldsymbol{x}|y)| = \left| \int p(\boldsymbol{x}|y') p_\kappa(y') dy' - p(\boldsymbol{x}|y) \right|$$

$$\leq \int |p(\boldsymbol{x}|y') - p(\boldsymbol{x}|y)| \, p_\kappa(y') dy'$$

(by (A2))

$$\leq \int g^r(\boldsymbol{x}) |y' - y| p_\kappa(y') dy'$$

$$\leq \kappa g^r(\boldsymbol{x}).$$

Thus, Eq. (S.18) is upper bounded as follows,

$$\sup_{D \in \mathcal{D}} \left| \mathbb{E}_{\boldsymbol{x} \sim p_r^{y,\kappa}(\boldsymbol{x})} \left[ -\log D(\boldsymbol{x}, y) \right] - \mathbb{E}_{\boldsymbol{x} \sim p_r(\boldsymbol{x}|y)} \left[ -\log D(\boldsymbol{x}, y) \right] \right|$$

$$\leq \int \kappa g^r(\boldsymbol{x}) d\boldsymbol{x}$$

(by (A2))

$$= \kappa M^r.$$

(S.19)

By combining Eq. (S.17) and (S.19), we can get Eq. (S.16), which finishes the proof. $\square$

Similarly, we apply identical proof strategy to the fake images $\boldsymbol{x}^g$ and generator distribution $p_g(\boldsymbol{x}|y)$.

**Lemma S.2.** *Suppose that (A1), (A3) and (A4) hold, then $\forall \delta \in (0,1)$, with probability at least $1 - \delta$,*

$$
\sup_{D \in \mathcal{D}} \left| \frac{1}{N_{y,\kappa}^g} \sum_{i=1}^{N^g} \mathbb{1}_{\{|y-y_i^g| \le \kappa\}} \left[ -\log(1 - D(\boldsymbol{x}_i, y)) \right] - \mathbb{E}_{\boldsymbol{x} \sim p_g(\boldsymbol{x}|y)} \left[ -\log(1 - D(\boldsymbol{x}, y)) \right] \right|
$$
$$
\le U \sqrt{\frac{1}{2N_{y,\kappa}^g} \log\left(\frac{2}{\delta}\right)} + \frac{\kappa U M^g}{2}, \tag{S.20}
$$

*for a given $y$.*

*Proof.* This proof is omitted because it is almost identical to the one for Lemma S.1. □

The following two lemmas provide the bounds for SVDL.

**Lemma S.3.** *Suppose that (A1), (A2) and (A4) hold, then $\forall \delta \in (0,1)$, with probability at least $1 - \delta$,*

$$
\sup_{D \in \mathcal{D}} \left| \frac{\frac{1}{N^r} \sum_{i=1}^{N^r} w^r(y_i^r, y) \left[ -\log D(\boldsymbol{x}_i^r, y) \right]}{\frac{1}{N^r} \sum_{i=1}^{N^r} w^r(y_i^r, y)} - \mathbb{E}_{\boldsymbol{x} \sim p_r(\boldsymbol{x}|y)} \left[ -\log D(\boldsymbol{x}, y) \right] \right|
$$
$$
\le \frac{U}{W^r(y)} \sqrt{\frac{1}{2N^r} \log\left(\frac{4}{\delta}\right)} + \frac{U M^r}{2} \mathbb{E}_{y' \sim p_w^r(y'|y)} \left[ |y' - y| \right], \tag{S.21}
$$

*for a given $y$.*

*Proof.* For brevity, denote by $f(x, y) = -\log D(x, y)$ and $\mathcal{F} = -\log \mathcal{D}$. Then,

$$
\sup_{D \in \mathcal{D}} \left| \frac{\frac{1}{N^r} \sum_{i=1}^{N^r} w^r(y_i^r, y) \left[ -\log D(\boldsymbol{x}_i^r, y) \right]}{\frac{1}{N^r} \sum_{i=1}^{N^r} w^r(y_i^r, y)} - \mathbb{E}_{\boldsymbol{x} \sim p_r(\boldsymbol{x}|y)} \left[ -\log D(\boldsymbol{x}, y) \right] \right|
$$
$$
= \sup_{f \in \mathcal{F}} \left| \frac{\frac{1}{N^r} \sum_{i=1}^{N^r} w^r(y_i^r, y) f(\boldsymbol{x}_i^r, y)}{\frac{1}{N^r} \sum_{i=1}^{N^r} w^r(y_i^r, y)} - \mathbb{E}_{\boldsymbol{x} \sim p_r(\boldsymbol{x}|y)} \left[ f(\boldsymbol{x}, y) \right] \right| \tag{S.22}
$$
$$
\le \sup_{f \in \mathcal{F}} \left| \frac{\frac{1}{N^r} \sum_{i=1}^{N^r} w^r(y_i^r, y) f(\boldsymbol{x}_i^r, y)}{\frac{1}{N^r} \sum_{i=1}^{N^r} w^r(y_i^r, y)} - \mathbb{E}_{\boldsymbol{x} \sim p_r^{y, w^r}(\boldsymbol{x})} \left[ f(\boldsymbol{x}, y) \right] \right|
$$
$$
+ \sup_{f \in \mathcal{F}} \left| \mathbb{E}_{\boldsymbol{x} \sim p_r^{y, w^r}(\boldsymbol{x})} \left[ f(\boldsymbol{x}, y) \right] - \mathbb{E}_{\boldsymbol{x} \sim p_r(\boldsymbol{x}|y)} \left[ f(\boldsymbol{x}, y) \right] \right|
$$

where the inequality is by triangular inequality. We then derive bounds for both two terms of the last line.

1. For the first term, we can further split it into two parts,

$$
\left| \frac{\frac{1}{N^r} \sum_{i=1}^{N^r} w^r(y_i^r, y) f(\boldsymbol{x}_i^r, y)}{\frac{1}{N^r} \sum_{i=1}^{N^r} w^r(y_i^r, y)} - \mathbb{E}_{\boldsymbol{x} \sim p_r^{y, w^r}(\boldsymbol{x})} \left[ f(\boldsymbol{x}, y) \right] \right|
$$
$$
\le \left| \frac{\frac{1}{N^r} \sum_{i=1}^{N^r} w^r(y_i^r, y) f(\boldsymbol{x}_i^r, y)}{\frac{1}{N^r} \sum_{i=1}^{N^r} w^r(y_i^r, y)} - \frac{\frac{1}{N^r} \sum_{i=1}^{N^r} w^r(y_i^r, y) f(\boldsymbol{x}_i^r, y)}{W^r(y)} \right| \tag{S.23}
$$
$$
+ \left| \frac{\frac{1}{N^r} \sum_{i=1}^{N^r} w^r(y_i^r, y) f(\boldsymbol{x}_i^r, y)}{W^r(y)} - \mathbb{E}_{\boldsymbol{x} \sim p_r^{y, w^r}(\boldsymbol{x})} \left[ f(\boldsymbol{x}, y) \right] \right|
$$

Focusing on the first part of RHS of Eq.(S.23). By (A1),

$$
\left| \frac{\frac{1}{N^r} \sum_{i=1}^{N^r} w^r(y_i^r, y) f(\boldsymbol{x}_i^r, y)}{\frac{1}{N^r} \sum_{i=1}^{N^r} w^r(y_i^r, y)} - \frac{\frac{1}{N^r} \sum_{i=1}^{N^r} w^r(y_i^r, y) f(\boldsymbol{x}_i^r, y)}{W^r(y)} \right|
$$
$$
\le U \frac{\left| \frac{1}{N^r} \sum_{i=1}^{N^r} w^r(y_i^r, y) - W^r(y) \right|}{W^r(y)}
$$

Note that $\forall y, y', w^r(y', y) = e^{-\nu|y-y'|^2} \leq 1$ and hence given $y$, $w^r(y', y)$ is a random variable bounded by 1. Apply Hoeffding's inequality to the numerator of above, yielding that with probability at least $1 - \delta'$,

$$\left| \frac{\frac{1}{N^r}\sum_{i=1}^{N^r} w^r(y_i^r, y)f(\boldsymbol{x}_i^r, y)}{\frac{1}{N^r}\sum_{i=1}^{N^r} w^r(y_i^r, y)} - \frac{\frac{1}{N^r}\sum_{i=1}^{N^r} w^r(y_i^r, y)f(\boldsymbol{x}_i^r, y)}{W^r(y)} \right| \leq \frac{U}{W^r(y)}\sqrt{\frac{1}{2N^r}\log\left(\frac{2}{\delta'}\right)}.$$

(S.24)

Then, consider the second part of RHS of Eq.(S.23). Recall that $p_r^{y,w^r}(\boldsymbol{x}) \triangleq \int p_r(\boldsymbol{x}|y')\frac{w^r(y',y)p^r(y')}{W^r(y)}dy'$. Thus,

$$\left| \frac{\frac{1}{N^r}\sum_{i=1}^{N^r} w^r(y_i^r, y)f(\boldsymbol{x}_i^r, y)}{W^r(y)} - \mathbb{E}_{\boldsymbol{x}\sim p_r^{y,w^r}(\boldsymbol{x})}[f(\boldsymbol{x}, y)] \right|$$

$$= \frac{1}{W^r(y)}\left| \frac{1}{N^r}\sum_{i=1}^{N^r} w^r(y_i^r, y)f(\boldsymbol{x}_i^r, y) - \mathbb{E}_{(\boldsymbol{x},y')\sim p_r(\boldsymbol{x},y')}[w^r(y', y)f(\boldsymbol{x}_i^r, y)] \right|,$$

where $p_r(\boldsymbol{x}, y') = p_r(\boldsymbol{x}|y')p^r(y')$ denotes the joint distribution of real image and its label. Again, since $w^r(y', y)f(\boldsymbol{x}_i^r, y)$ is uniformly bounded by $U$ under (A1), we can apply Hoeffding's inequality. This implies that with probability at least $1 - \delta'$, the above can be upper bounded by

$$\frac{U}{W^r(y)}\sqrt{\frac{1}{2N^r}\log\left(\frac{2}{\delta'}\right)}.$$

(S.25)

Combining Eq. (S.24) and (S.25) and by setting $\delta' = \frac{\delta}{2}$, we have with probability at least $1 - \delta$,

$$\left| \frac{\frac{1}{N^r}\sum_{i=1}^{N^r} w^r(y_i^r, y)f(\boldsymbol{x}_i^r, y)}{\frac{1}{N^r}\sum_{i=1}^{N^r} w^r(y_i^r, y)} - \mathbb{E}_{\boldsymbol{x}\sim p_r^{y,w^r}(\boldsymbol{x})}[f(\boldsymbol{x}, y)] \right| \leq \frac{U}{W^r(y)}\sqrt{\frac{1}{2N^r}\log\left(\frac{4}{\delta}\right)}.$$

Since this holds for $\forall f \in \mathcal{F}$, taking supremum over $f$, we have

$$\sup_{f\in\mathcal{F}}\left| \frac{\frac{1}{N^r}\sum_{i=1}^{N^r} w^r(y_i^r, y)f(\boldsymbol{x}_i^r, y)}{\frac{1}{N^r}\sum_{i=1}^{N^r} w^r(y_i^r, y)} - \mathbb{E}_{\boldsymbol{x}\sim p_r^{y,w^r}(\boldsymbol{x})}[f(\boldsymbol{x}, y)] \right| \leq \frac{U}{W^r(y)}\sqrt{\frac{1}{2N^r}\log\left(\frac{4}{\delta}\right)}.$$

(S.26)

2. For the second term on the RHS of Eq.(S.22). By (A1) that $|f| < U$,

$$\sup_{f\in\mathcal{F}}\left| \mathbb{E}_{\boldsymbol{x}\sim p_r^{y,w^r}(\boldsymbol{x})}[f(\boldsymbol{x}, y)] - \mathbb{E}_{\boldsymbol{x}\sim p_r(\boldsymbol{x}|y)}[f(\boldsymbol{x}, y)] \right|$$

$$\leq U\|p_r^{y,w^r}(\boldsymbol{x}) - p_r(\boldsymbol{x}|y)\|_{TV}$$

$$= \frac{U}{2}\int |p_r^{y,w^r}(\boldsymbol{x}) - p_r(\boldsymbol{x}|y)|d\boldsymbol{x}.$$

Note that by the definition of $p_r^{y,w^r}(\boldsymbol{x}) \triangleq \int p_r(\boldsymbol{x}|y')\frac{w^r(y',y)p^r(y')}{W^r(y)}dy'$ and $p_w^r(y'|y) \triangleq \frac{w^r(y',y)p^r(y')}{W^r(y)}$, we have

$$|p_r^{y,w^r}(\boldsymbol{x}) - p_r(\boldsymbol{x}|y)| = \left| \int p_r(\boldsymbol{x}|y')p_w^r(y'|y)\,dy' - p_r(\boldsymbol{x}|y) \right|$$

$$\leq \int |p_r(\boldsymbol{x}|y') - p_r(\boldsymbol{x}|y)|\,p_w^r(y'|y)\,dy'.$$

By (A.2) and $y \in [0, 1]$, the above is upper bounded by $g^r(\boldsymbol{x})\mathbb{E}_{y'\sim p_w^r(y'|y)}[|y - y'|]$. Thus,

$$\sup_{f\in\mathcal{F}}\left| \mathbb{E}_{\boldsymbol{x}\sim p_r^{y,w^r}(\boldsymbol{x})}[f(\boldsymbol{x}, y)] - \mathbb{E}_{\boldsymbol{x}\sim p_r(\boldsymbol{x}|y)}[f(\boldsymbol{x}, y)] \right|$$

$$\leq \frac{U}{2}\int g^r(\boldsymbol{x})\mathbb{E}_{y'\sim p_w^r(y'|y)}[|y' - y|]\,d\boldsymbol{x}$$

(S.27)

$$= \frac{UM^r}{2}\mathbb{E}_{y'\sim p_w^r(y'|y)}[|y' - y|].$$

Therefore, combining both Eq.(S.26) and (S.27), with probability at least $1 - \delta$,

$$\sup_{D \in \mathcal{D}} \left| \frac{\frac{1}{N^r} \sum_{i=1}^{N^r} w^r(y_i^r, y) \left[ -\log D(\boldsymbol{x}_i^r, y) \right]}{\frac{1}{N^r} \sum_{i=1}^{N^r} w^r(y_i^r, y)} - \mathbb{E}_{\boldsymbol{x} \sim p_r(\boldsymbol{x}|y)} \left[ -\log D(\boldsymbol{x}, y) \right] \right|$$

$$\leq \frac{U}{W^r(y)} \sqrt{\frac{1}{2N^r} \log \left( \frac{4}{\delta} \right)} + \frac{U M^r}{2} \mathbb{E}_{y' \sim p_w^r(y'|y)} \left[ |y' - y| \right].$$

This finishes the proof.

$\square$

**Lemma S.4.** *Suppose that (A1), (A3) and (A4) hold, then $\forall \delta \in (0, 1)$, with probability at least $1 - \delta$,*

$$\sup_{D \in \mathcal{D}} \left| \frac{\frac{1}{N^g} \sum_{i=1}^{N^g} w^g(y_i^g, y) \left[ -\log(1 - D(\boldsymbol{x}_i^g, y)) \right]}{\frac{1}{N^g} \sum_{i=1}^{N^g} w^g(y_i^g, y)} - \mathbb{E}_{\boldsymbol{x} \sim p_g(\boldsymbol{x}|y)} \left[ -\log(1 - D(\boldsymbol{x}, y)) \right] \right| \tag{S.28}$$

$$\leq \frac{U}{W^g(y)} \sqrt{\frac{1}{2N^g} \log \left( \frac{4}{\delta} \right)} + \frac{U M^g}{2} \mathbb{E}_{y' \sim p_w^g(y'|y)} \left[ |y' - y| \right],$$

*for a given $y$.*

*Proof.* This proof is omitted because it is almost identical to the one for Lemma S.21. $\square$

As introduced in Section 2, we use KDE for the marginal label distribution with Gaussian kernel. The next theorem characterizes the difference between a $p_r(y), p_g(y)$ and their KDE using $n$ i.i.d. samples.

**Theorem S.3.** *Let $\hat{p}_r^{KDE}(y)$ and $\hat{p}_g^{KDE}(y)$ stand for the KDE of $p_r(y)$ and $p_g(y)$ respectively. Under condition (A4), if the KDEs are based on $n$ i.i.d. samples from $p_r/p_g$ and a bandwidth $\sigma$, for all $\delta \in (0, 1)$, with probability at least $1 - \delta$,*

$$\sup_t \left| \hat{p}_r^{KDE}(y) - p_r(y) \right| \leq \sqrt{\frac{C_{1,\delta}^{KDE} \log n}{n\sigma}} + L^r \sigma^2, \tag{S.29}$$

$$\sup_t \left| \hat{p}_g^{KDE}(y) - p_g(y) \right| \leq \sqrt{\frac{C_{2,\delta}^{KDE} \log n}{n\sigma}} + L^g \sigma^2, \tag{S.30}$$

*for some constants $C_{1,\delta}^{KDE}, C_{2,\delta}^{KDE}$ depending on $\delta$.*

*Proof.* By ((Wasserman); P.12), for any $p(t) \in \Sigma(L)$ (the Hölder Class, see Definition 1), with probability at least $1 - \delta$,

$$\sup_t \left| \hat{p}^{KDE}(t) - p(t) \right| \leq \sqrt{\frac{C_{\delta}^{KDE} \log n}{n\sigma}} + c\sigma^2, \tag{S.31}$$

for some constants $C_{\delta}^{KDE}$ and $c$, where $C$ depends on $\delta$ and $c = L \int K(s)|s|^2 ds$. Since in this work, $K$ is chosen as Gaussian kernel, $c = L \int K(s)|s|^2 ds = L$. $\square$

### S.5.2.2 ERROR BOUNDS FOR HVDL AND SVDL

Based on the lemmas and theorems in Supp. S.5.2.1, we derive the error bounds of HVDL and SVDL, which will be used in the proofs of Theorems 1 and 2.

**Theorem S.4.** *Assume that (A1)-(A4) hold, then $\forall \delta \in (0, 1)$, with probability at least $1 - \delta$,*

$$\sup_{D \in \mathcal{D}} \left| \widehat{\mathcal{L}}^{HVDL}(D) - \mathcal{L}(D) \right|$$

$$\leq U \left( \sqrt{\frac{C_{1,\delta}^{KDE} \log N^r}{N^r \sigma}} + L^r \sigma^2 \right) + U \left( \sqrt{\frac{C_{2,\delta}^{KDE} \log N^g}{N^g \sigma}} + L^g \sigma^2 \right) + \frac{\kappa U (M^r + M^g)}{2} \tag{S.32}$$

$$+ U \sqrt{\frac{1}{2} \log \left( \frac{8}{\delta} \right)} \left( \mathbb{E}_{y \sim \hat{p}_r^{KDE}(y)} \left[ \sqrt{\frac{1}{N_{y,\kappa}^r}} \right] + \mathbb{E}_{y \sim \hat{p}_g^{KDE}(y)} \left[ \sqrt{\frac{1}{N_{y,\kappa}^g}} \right] \right),$$

*for some constants $C_{1,\delta}^{KDE}, C_{2,\delta}^{KDE}$ depending on $\delta$.*

*Proof.* We first decompose $\sup_{D \in \mathcal{D}} \left| \widehat{\mathcal{L}}^{\text{HVDL}}(D) - \mathcal{L}(D) \right|$ as follows

$$
\sup_{D \in \mathcal{D}} \left| \widehat{\mathcal{L}}^{\text{HVDL}}(D) - \mathcal{L}(D) \right|
$$

$$
\leq \sup_{D \in \mathcal{D}} \left| \int \left[ \int \left[ -\log D(\boldsymbol{x}, y) \right] p_r(\boldsymbol{x}|y) d\boldsymbol{x} \right] (p_r(y) - \hat{p}_r^{\text{KDE}}(y)) dy \right|
$$

$$
+ \sup_{D \in \mathcal{D}} \left| \int \left[ \int \left[ -\log(1 - D(\boldsymbol{x}, y)) \right] p_g(\boldsymbol{x}|y) d\boldsymbol{x} \right] (p_g(y) - \hat{p}_g^{\text{KDE}}(y)) dy \right|
$$

$$
+ \sup_{D \in \mathcal{D}} \left| \int \left[ \frac{1}{N_{y,\kappa}^r} \sum_{i=1}^{N^r} \mathbb{1}_{\{|y - y_i^r| \leq \kappa\}} \left[ -\log D(\boldsymbol{x}_i^r, y) \right] - \mathbb{E}_{\boldsymbol{x} \sim p_r(\boldsymbol{x}|y)} \left[ -\log D(\boldsymbol{x}, y) \right] \right] \hat{p}_r^{\text{KDE}}(y) dy \right|
$$

$$
+ \sup_{D \in \mathcal{D}} \left| \int \left[ \frac{1}{N_{y,\kappa}^g} \sum_{i=1}^{N^r} \mathbb{1}_{\{|y - y_i^g| \leq \kappa\}} \left[ -\log(1 - D(\boldsymbol{x}_i^g, y)) \right] - \mathbb{E}_{\boldsymbol{x} \sim p_g(\boldsymbol{x}|y)} \left[ -\log(1 - D(\boldsymbol{x}, y)) \right] \right] \hat{p}_g^{\text{KDE}}(y) dy \right|.
$$

These four terms in the RHS can be bounded separately as follows

1. The first term can be bounded by using Theorem S.3 and the boundness of $D$ and $y \in [0, 1]$. For the first term, $\forall \delta_1 \in (0, 1)$, with at least probability $1 - \delta_1$,

$$
\sup_{D \in \mathcal{D}} \left| \int \left[ \int \left[ -\log D(\boldsymbol{x}, y) \right] p_r(\boldsymbol{x}|y) d\boldsymbol{x} \right] (p_r(y) - \hat{p}_r^{\text{KDE}}(y)) dy \right|
$$

$$
\leq U \left( \sqrt{\frac{C_{1,\delta_1}^{\text{KDE}} \log N^r}{N^r \sigma}} + L^r \sigma^2 \right), \tag{S.33}
$$

for some constants $C_{1,\delta_1}^{\text{KDE}}$ depending on $\delta_1$.

2. The second term can be bounded by using Theorem S.3 and the boundness of $D$ and $y \in [0, 1]$. For the first term, $\forall \delta_2 \in (0, 1)$, with at least probability $1 - \delta_2$,

$$
\sup_{D \in \mathcal{D}} \left| \int \left[ \int \left[ -\log(1 - D(\boldsymbol{x}, y)) \right] p_g(\boldsymbol{x}|y) d\boldsymbol{x} \right] (p_g(y) - \hat{p}_g^{\text{KDE}}(y)) dy \right|
$$

$$
\leq U \left( \sqrt{\frac{C_{2,\delta_2}^{\text{KDE}} \log N^g}{N^r \sigma}} + L^g \sigma^2 \right), \tag{S.34}
$$

for some constants $C_{2,\delta_2}^{\text{KDE}}$ depending on $\delta_2$.

3. The third term can be bounded by using Lemma S.1 and S.2. For the third term, $\forall \delta_3 \in (0, 1)$, with at least probability $1 - \delta_3$,

$$
\sup_{D \in \mathcal{D}} \left| \int \left[ \frac{1}{N_{y,\kappa}^r} \sum_{i=1}^{N^r} \mathbb{1}_{\{|y - y_i^r| \leq \kappa\}} \left[ -\log D(\boldsymbol{x}_i^r, y) \right] - \mathbb{E}_{\boldsymbol{x} \sim p_r(\boldsymbol{x}|y)} \left[ -\log D(\boldsymbol{x}, y) \right] \right] \hat{p}_r^{\text{KDE}}(y) dy \right|
$$

$$
\leq \int \sup_{D \in \mathcal{D}} \left| \frac{1}{N_{y,\kappa}^r} \sum_{i=1}^{N^r} \mathbb{1}_{\{|y - y_i^r| \leq \kappa\}} \left[ -\log D(\boldsymbol{x}_i^r, y) \right] - \mathbb{E}_{\boldsymbol{x} \sim p_r(\boldsymbol{x}|y)} \left[ -\log D(\boldsymbol{x}, y) \right] \right| \hat{p}_r^{\text{KDE}}(y) dy
$$

$$
\leq \int \left[ U \sqrt{\frac{1}{2N_{y,\kappa}^r} \log \left( \frac{2}{\delta_3} \right)} + \frac{\kappa U M^r}{2} \right] \hat{p}_r^{\text{KDE}}(y) dy
$$

Note that $N_{y,\kappa}^r = \sum_{i=1}^{N^r} \mathbb{1}_{\{|y-y_i^r|\}}$, which is a random variable of $y_i's$. The above can be expressed as

$$\sup_{D \in \mathcal{D}} \left| \int \left[ \frac{1}{N_{y,\kappa}^r} \sum_{i=1}^{N^r} \mathbb{1}_{\{|y-y_i^r|\leq\kappa\}} \left[ -\log D(\boldsymbol{x}_i^r, y) \right] - \mathbb{E}_{\boldsymbol{x}\sim p_r(\boldsymbol{x}|y)} \left[ -\log D(\boldsymbol{x}, y) \right] \right] \hat{p}_r^{\mathrm{KDE}}(y) dy \right|$$

$$\leq \frac{\kappa U M^r}{2} + U \sqrt{\frac{1}{2} \log\left(\frac{2}{\delta_3}\right)} \mathbb{E}_{y\sim\hat{p}_r^{\mathrm{KDE}}(y)} \left[ \sqrt{\frac{1}{N_{y,\kappa}^r}} \right].$$

$$(\text{S.35})$$

4. Similarly, for the fourth term, $\forall \delta_4 \in (0,1)$, with at least probability $1 - \delta_4$,

$$\sup_{D \in \mathcal{D}} \left| \int \left\{ \int \left[ \frac{1}{N_{y,\kappa}^g} \sum_{i=1}^{N^r} \mathbb{1}_{\{|y-y_i^g|\leq\kappa\}} \left[ -\log(1 - D(\boldsymbol{x}_i^g, y)) \right] \right. \right. \right.$$

$$\left. \left. \left. - \mathbb{E}_{\boldsymbol{x}\sim p_g(\boldsymbol{x}|y)} \left[ -\log(1 - D(\boldsymbol{x}, y)) \right] \right] d\boldsymbol{x} \right\} \hat{p}_g^{\mathrm{KDE}}(y) dy \right| \qquad (\text{S.36})$$

$$\leq \frac{\kappa U M^g}{2} + U \sqrt{\frac{1}{2} \log\left(\frac{2}{\delta_4}\right)} \mathbb{E}_{y\sim\hat{p}_g^{\mathrm{KDE}}(y)} \left[ \sqrt{\frac{1}{N_{y,\kappa}^g}} \right].$$

With $\delta_1 = \delta_2 = \delta_3 = \delta_4 = \frac{\delta}{4}$, combining Eq. (S.33) - (S.36) leads to the upper bound in Theorem S.4. $\qquad \square$

**Theorem S.5.** *Assume that (A1)-(A4) hold, then $\forall \delta \in (0,1)$, with probability at least $1 - \delta$,*

$$\sup_{D \in \mathcal{D}} \left| \widehat{\mathcal{L}}^{SVDL}(D) - \mathcal{L}(D) \right|$$

$$\leq U \left( \sqrt{\frac{C_{1,\delta}^{KDE} \log N^r}{N^r \sigma}} + L^r \sigma^2 \right) + U \left( \sqrt{\frac{C_{2,\delta}^{KDE} \log N^g}{N^g \sigma}} + L^g \sigma^2 \right)$$

$$+ U \sqrt{\frac{1}{2} \log\left(\frac{16}{\delta}\right)} \left( \frac{1}{\sqrt{N^r}} \mathbb{E}_{y\sim\hat{p}_r^{KDE}(y)} \left[ \frac{1}{W^r(y)} \right] + \frac{1}{\sqrt{N^g}} \mathbb{E}_{y\sim\hat{p}_g^{KDE}(y)} \left[ \frac{1}{W^g(y)} \right] \right) \qquad (\text{S.37})$$

$$+ \frac{U}{2} \left( M^r \mathbb{E}_{y\sim\hat{p}_r^{KDE}(y)} \left[ \mathbb{E}_{y'\sim p_w^r(y'|y)} |y' - y| \right] + M^g \mathbb{E}_{y\sim\hat{p}_g^{KDE}(y)} \left[ \mathbb{E}_{y'\sim p_w^g(y'|y)} |y' - y| \right] \right)$$

*for some constant $C_{1,\delta}^{KDE}, C_{2,\delta}^{KDE}$ depending on $\delta$.*

*Proof.* Similar to the decomposition for Theorem S.4, we can decompose $\sup_{D \in \mathcal{D}} \left| \widehat{\mathcal{L}}^{\mathrm{SVDL}}(D) - \mathcal{L}(D) \right|$ into four terms which can be bounded by using Theorem S.3, the boundness of $D$, Lemma S.3, and Lemma S.4. The detail is omitted because it is almost identical to the one of Theorem S.4. $\qquad \square$

### S.5.2.3 PROOF OF THEOREM 1

Based on Theorem S.4, we derive Theorem 1.

*Proof.* We first decompose $\mathcal{L}(\widehat{D}^{\mathrm{HVDL}}) - \mathcal{L}(D^*)$ as follows

$$
\begin{aligned}
&\mathcal{L}(\widehat{D}^{\mathrm{HVDL}}) - \mathcal{L}(D^*) \\
=&\mathcal{L}(\widehat{D}^{\mathrm{HVDL}}) - \widehat{\mathcal{L}}(\widehat{D}^{\mathrm{HVDL}}) + \widehat{\mathcal{L}}(\widehat{D}^{\mathrm{HVDL}}) - \widehat{\mathcal{L}}(\widetilde{D}) + \widehat{\mathcal{L}}(\widetilde{D}) - \mathcal{L}(\widetilde{D}) + \mathcal{L}(\widetilde{D}) - \mathcal{L}(D^*) \\
&\text{(by } \widehat{\mathcal{L}}(\widehat{D}^{\mathrm{HVDL}}) - \widehat{\mathcal{L}}(\widetilde{D}) \le 0) \\
\le & 2 \sup_{D \in \mathcal{D}} \left| \widehat{\mathcal{L}}^{\mathrm{HVDL}}(D) - \mathcal{L}(D) \right| + \mathcal{L}(\widetilde{D}) - \mathcal{L}(D^*) \\
&\text{(by Theorem S.4)} \\
\le & 2U \left( \sqrt{\frac{C_{1,\delta}^{\mathrm{KDE}} \log N^r}{N^r \sigma}} + L^r \sigma^2 \right) + 2U \left( \sqrt{\frac{C_{2,\delta}^{\mathrm{KDE}} \log N^g}{N^g \sigma}} + L^g \sigma^2 \right) + \kappa U(M^r + M^g) \\
&+ 2U \sqrt{\frac{1}{2} \log\left(\frac{8}{\delta}\right)} \left( \mathbb{E}_{y \sim \hat{p}_r^{\mathrm{KDE}}(y)} \left[ \sqrt{\frac{1}{N_{y,\kappa}^r}} \right] + \mathbb{E}_{y \sim \hat{p}_g^{\mathrm{KDE}}(y)} \left[ \sqrt{\frac{1}{N_{y,\kappa}^g}} \right] \right) + \mathcal{L}(\widetilde{D}) - \mathcal{L}(D^*).
\end{aligned}
$$
(S.38)

$\square$

### S.5.2.4 PROOF OF THEOREM 2

Based on Theorem S.5, we derive Theorem 2.

*Proof.* The detail is omitted because it is almost identical to the one of Theorem 1 in Supp. S.5.2.3.

$\square$

### S.5.2.5 INTERPRETATION OF THEOREMS 1 AND 2

Both theorems imply HVDL and SVDL perform well if the output of $D$ is not too close to 0 or 1 (i.e., favor small $U$). The first two terms in both upper bounds control the quality of KDE, which implies KDE works better if we have larger $N^r$ and $N^g$ and a smaller $\sigma$. The rest terms of the two bounds are different. In the HVDL case, we favor smaller $\kappa$, $M^r$, and $M^g$. However, we should avoid setting $\kappa$ for a too small value because we prefer larger $N_{y,\kappa}^r$ and $N_{y,\kappa}^g$. In the SVDL case, we prefer small $M^r$ and $M^g$ but large $W^r(y)$ and $W^g(y)$. Large $W^r(y)$ and $W^g(y)$ imply that the weight function decays slowly (i.e., small $\nu$; similar to large $N_{y,\kappa}^r$ and $N_{y,\kappa}^g$ in Eq.(S.32)). However, we should avoid setting $\nu$ too small because a small $\nu$ leads to large $\mathbb{E}_{y' \sim \hat{p}_w^r(y'|y)} |y' - y|$ and $\mathbb{E}_{y' \sim \hat{p}_w^g(y'|y)} |y' - y|$ (i.e., $y'$'s which are far away from $y$ have large weights). In our experiments, we use some rule-of-thumb formulae to select $\kappa$ and $\nu$. As a future work, a refined hyper-parameter selection method should be proposed.

## S.6 MORE DETAILS OF THE SIMULATION IN SECTION 4.1

### S.6.1 NETWORK ARCHITECTURES

Please refer to Table S.6.1 and Table S.6.2 for the network architectures we adopted for cGAN and CcGAN in our Simulation experiments.

### S.6.2 TRAINING SETUPS

The cGAN and CcGAN are trained for 6000 iterations on the training set with the Adam (Kingma & Ba, 2015) optimizer (with $\beta_1 = 0.5$ and $\beta_2 = 0.999$), a constant learning rate $5 \times 10^{-5}$ and batch size 128. The rule of thumb formulae in Section S.4 are used to select the hyper-parameters for HVDL and SVDL, where we let $m_\kappa = 2$. Thus, the three hyper-parameters in this experiments are set as follows: $\sigma = 0.074$, $\kappa = 0.017$, $\nu = 3600$.

Table S.6.1: Network architectures for the generator and discriminator of **cGAN** in the simulation. "fc" denotes a fully-connected layer. "BN" stands for batch normalization. The label $y$ is treated as a class label and encoded by label-embedding (Akata et al., 2015) so its dimension equals to the number of distinct angles in the training set (i.e., $y \in \mathbb{R}^{120}$).

| (a) Generator | (b) Discriminator |
|---|---|
| $z \in \mathbb{R}^2 \sim N(0, I); y \in \mathbb{R}^{120}$ | A sample $x \in \mathbb{R}^2$ |
| concat$(z, y) \in \mathbb{R}^{122}$ | fc$\to 100$; ReLU |
| fc$\to 100$; BN; ReLU | fc$\to 100$; ReLU |
| fc$\to 100$; BN; ReLU | fc$\to 100$; ReLU |
| fc$\to 100$; BN; ReLU | fc$\to 100$; ReLU |
| fc$\to 100$; BN; ReLU | concat(output of previous layer, $y$) $\in \mathbb{R}^{220}$, |
| fc$\to 100$; BN; ReLU | where $y \in \mathbb{R}^{120}$ is the label of $x$. |
| fc$\to 100$; BN; ReLU | fc$\to 100$; ReLU |
| fc$\to 2$ | fc$\to 1$; Sigmoid |

Table S.6.2: Network architectures for the generator and discriminator of our proposed **CcGAN** in the simulation. The label $y$ is treated as a real scalar so its dimension is 1. We do not directly input $y$ into the generator and discriminator. We first convert each $y$ into the coordinate of the mean represented by this $y$, i.e., $(\sin(y), \cos(y))$. Then we insert this coordinate into the networks.

| (a) Generator | (b) Discriminator |
|---|---|
| $z \in \mathbb{R}^2 \sim N(0, I); y \in \mathbb{R}$ | A sample $x \in \mathbb{R}^2$ with label $y \in \mathbb{R}$ |
| concat$(z, \sin(y), \cos(y)) \in \mathbb{R}^4$ | concat$(x, \sin(y), \cos(y)) \in \mathbb{R}^4$ |
| fc$\to 100$; BN; ReLU | fc$\to 100$; ReLU |
| fc$\to 100$; BN; ReLU | fc$\to 100$; ReLU |
| fc$\to 100$; BN; ReLU | fc$\to 100$; ReLU |
| fc$\to 100$; BN; ReLU | fc$\to 100$; ReLU |
| fc$\to 100$; BN; ReLU | fc$\to 100$; ReLU |
| fc$\to 100$; BN; ReLU | fc$\to 1$; Sigmoid |
| fc$\to 2$ | |

### S.6.3 TESTING SETUPS

When evaluating the trained cGAN, if a test label $y'$ is unseen in the training set, we first find its closest, seen label $y$. Then, we generate samples from the trained cGAN at $y$ instead of at $y'$. On the contrary, generating samples from CcGAN at unseen labels is well-defined.

### S.6.4 EXTRA EXPERIMENTS

#### S.6.4.1 VARYING NUMBER OF GAUSSIANS FOR TRAINING DATA GENERATION

In this section, we study the influence of the number of Gaussians used for training data generation on the performance of cGAN and CcGAN. We vary the number of Gaussians from 120 to 10 with step size 10 but keep other settings in Section 4.1 unchanged and plot the line graphs of 2-Wasserstein Distance (log scale) versus the number of Gaussians in Fig. S.6.1. Reducing the number of Gaussians for training implies a larger gap between any two consecutive distinct angles in the training set. **As the number of Gaussians decreases, the continuous scenario gradually degenerates to the categorical scenario, therefore the assumption that a small perturbation to $y$ results in a negligible change to $p(\boldsymbol{x}|y)$ is no longer satisfied.** Consequently, the 2-Wasserstein distances of the proposed two CcGAN methods gradually increase and eventually surpass the 2-Wasserstein distance of cGAN when the number of Gaussians is small (e.g., less than 40). Note that reducing the number of Gaussians in the training data generation will not improve the performance of cGAN in the testing

because many angles seen in the testing stage (we evaluate each method on 360 angles) do not appear in the training set.

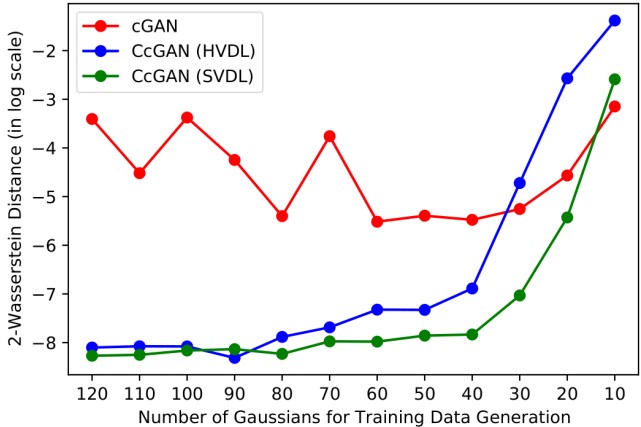

Figure S.6.1: Line graphs of 2-Wasserstein Distance (log scale) versus the number of Gaussians for training data generation. As the number of Gaussians decreases, the continuous scenario gradually degenerates to the categorical scenario, therefore the assumption that a small perturbation to $y$ results in a negligible change to $p(\boldsymbol{x}|y)$ is no longer satisfied. Consequently, the 2-Wasserstein distances of two CcGAN methods gradually increase and eventually surpass the 2-Wasserstein distance of cGAN when the number of Gaussians is small (e.g., less than 40).

## S.7   MORE DETAILS OF THE EXPERIMENT ON RC-49 IN SECTION 4.2

### S.7.1   DESCRIPTION OF RC-49

To generate RC-49, firstly we randomly select 49 3-D chair object models from the "Chair" category provided by ShapeNet (Chang et al., 2015). Then we use Blender v2.79 [1] to render these 3-D models. Specifically, during the rendering, we rotate each chair model along with the yaw axis for a degree between $0.1°$ and $89.9°$ (angle resolution as $0.1°$) where we use the scene image mode to compose our dataset. The rendered images are converted from the RGBA to RGB color model. In total, the RC-49 dataset consists of 44051 images of image size $64 \times 64$ in the PNG format.

### S.7.2   NETWORK ARCHITECTURES

The RC-49 dataset is a more sophisticated dataset compared with the simulation, thus it requires networks with deeper layers. We employ the SNGAN architecture (Miyato et al., 2018) in both cGAN and CcGAN consisting of residual blocks for the generator and the discriminator. Moreover, for the generator in cGAN, the regression labels are input into the network by the label embedding (Akata et al., 2015) and the conditional batch normalization (De Vries et al., 2017). For the discriminator in cGAN, the regression labels are fed into the network by the label embedding and the label projection (Miyato & Koyama, 2018). For CcGAN, the regression labels are fed into networks by our proposed label input method in Section 2. Please refer to our codes for more details about the network specifications of cGAN and CcGAN.

### S.7.3   TRAINING SETUPS

The cGAN and CcGAN are trained for 30,000 iterations on the training set with the Adam (Kingma & Ba, 2015) optimizer (with $\beta_1 = 0.5$ and $\beta_2 = 0.999$), a constant learning rate $10^{-4}$ and batch size 256. The rule of thumb formulae in Section S.4 are used to select the hyper-parameters for HVDL and SVDL, where we let $m_\kappa = 2$. Thus, the three hyper-parameters in this experiments are set as follows: $\sigma = 0.0473$, $\kappa = 0.004$, $\nu = 50625$.

---

[1]https://www.blender.org/download/releases/2-79/

## S.7.4 Testing setups

The RC-49 dataset consists of 899 distinct yaw angles and at each angle there are 49 images (corresponding to 49 types of chairs). At the test stage, we ask the trained cGAN or CcGAN to generate 200 fake images at each of these 899 yaw angles. Please note that, among these 899 yaw angles, only 450 of them are seen at the training stage so real images at the rest 449 angles are not used in the training.

We evaluate the quality of the fake images from three perspectives, i.e., visual quality, intra-label diversity, and label consistency. One overall metric (Intra-FID) and three separate metrics (NIQE, Diversity, and Label Score) are used. Their details are shown in Supp. S.7.5.

## S.7.5 Performance measures

Before we conduct the evaluation in terms of the four metrics, we first train an autoencoder (AE) , a regression CNN and a classification CNN on all real images in RC-49. The bottleneck dimension of the AE is 512 and the AE is trained to reconstruct the real images in RC-49 with MSE as the loss function. The regression CNN is trained to predict the yaw angle of a given image. The classification CNN is trained to predict the chair type of a given image. The autoencoder and both two CNNs are trained for 200 epochs with a batch size 256.

- **Intra-FID** (Miyato & Koyama, 2018): *We take Intra-FID as the overall score to evaluate the quality of fake images and we prefer the small Intra-FID score.* At each evaluation angle, we compute the FID (Heusel et al., 2017) between 49 real images and 200 fake images in terms of the bottleneck feature of the pre-trained AE. The Intra-FID score is the average FID over all 899 evaluation angles. Please note that we also try to use the classification CNN to compute the Intra-FID but the Intra-FID scores vary in a very wide range and sometimes obviously contradict with the three separate metrics.

- **NIQE** (Mittal et al., 2012): *NIQE is used to evaluate the visual quality of fake images with the real images as the reference and we prefer the small NIQE score.* We train one NIQE model with the 49 real images at each of the 899 angles so we have 899 NIQE models. During evaluation, a NIQE score is computed for each evaluation angle based on the NIQE model at that angle. Finally, we report the average and standard deviations of the 899 NIQE scores over the 899 yaw angels. Note that the NIQE is implemented by the NIQE module in `MATLAB`.

- **Diversity**: *Diversity is used to evaluate the intra-label diversity and the larger the better.* In RC-49, there are 49 chair types. At each evaluation angle, we ask the pre-trained classification to predict the chair types of the 200 fake images and an entropy is computed based on these predicted chair types. The diversity reported in Table 2 is the average of the 899 entropies over all evaluation angles.

- **Label Score**: *Label Score is used to evaluate the label consistency and the smaller the better.* We ask the pre-trained regression CNN to predict the yaw angles of all fake images and the predicted angles are then compared with the assigned angles. The Label Score is defined as the average absolute distance between the predicted angles and assigned angles over all fake images, which is equivalent to the Mean Absolute Error (MAE).

## S.7.6 Extra experiments

### S.7.6.1 More line graphs

### S.7.6.2 Interpolation

In Fig. S.7.3, we present some interpolation results of the two CcGAN methods (i.e., HVDL and SVDL). For an input pair $(z, y)$, we fix the noise $z$ but perform label-wise interpolations, i.e., varying label $y$ from 4.5 to 85.5. Clearly, all generated images are visually realistic and we can see the chair distribution smoothly changes over continuous angles. Please note that, Fig. S.7.3 is meant to show the smooth change of the chair distribution instead of one single chair so the chair type may change over angles. This confirms CcGAN is capable of capturing the underlying conditional image distribution rather than simply memorizing training data.

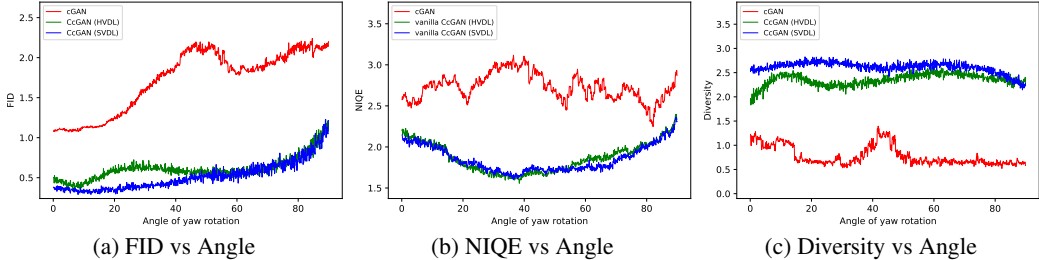

| (a) FID vs Angle | (b) NIQE vs Angle | (c) Diversity vs Angle |

Figure S.7.2: Line graphs of FID/NIQE/Diversity versus yaw angles on RC-49. Figs. S.7.2(a) to S.7.2(c) show that two CcGANs consistently outperform cGAN across all angles. The graphs of CcGANs also appear smoother than those of cGAN because of HVDL and SVDL.

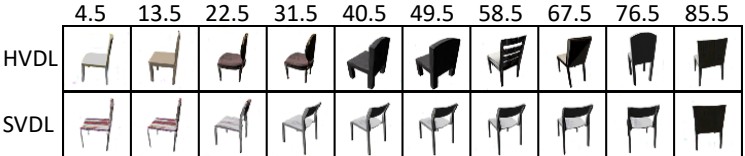

Figure S.7.3: Some example RC-49 fake images from the two CcGAN methods. We fix the noise $z$ but vary the label $y$.

### S.7.6.3 DEGENERATED CCGAN

In this experiment, we consider the extreme case of the proposed CcGAN (degenerated CcGAN), i.e., $\sigma \to 0$ and $\kappa \to 0$ or $\nu \to +\infty$. Then we train the degenerated CcGAN with the same experimental setting as for CcGANs. Some examples from degenerated CcGANs are shown in Fig. S.7.4. Since, at each angle, the degenerated CcGAN only uses the images at this angle, it leads to the mode collapse problem (e.g, the row in the yellow rectangle) and bad visual quality (e.g., images in the red rectangle) at some angles.

Note that the degenerated CcGAN is still different from cGAN, since we still treat $y$ as a continuous scalar instead of a class label here and we use the proposed label input method to incorporate $y$ into the generator and the discriminator.

### S.7.6.4 CGAN: DIFFERENT NUMBER OF CLASSES

In this experiment, we show that cGAN still fails even though we bin $[0.1, 89.9]$ into other number of classes. We experimented with three different bin setting – grouping labels into 90 classes, 150 classes and 210 classes, respectively. Experimental results are shown in Fig. S.7.5 and we observe that all three cGANs fail.

### S.7.6.5 VARYING SAMPLE SIZE FOR EACH DISTINCT ANGLE

To test cGAN and CcGAN under more challenging scenarios, we vary the sample size for each distinct angle in the training set from 45 to 5. We visualize the line graphs of Intra-FID versus the sample size for each distinct training angle in Fig. S.7.6. From this figure, we can see the two CcGAN methods substantially outperform cGAN no matter what is the sample size for each distinct angle in the training set. The overall trend in this figure also shows that smaller sample size reduces the performance of both cGAN and CcGAN.

### S.7.6.6 VARYING THE STRENGTH OF THE CORRELATION BETWEEN THE IMAGE AND ITS LABEL

To study how the strength of the correlation between the image $x$ and its label $y$ (i.e., the label power) influences the performance of cGAN and CcGAN, in this study, we randomly add Gaussian noises with a preset standard deviation to the raw regression labels in the training set. The strength of the correlation is controlled by the standard deviation of the Gaussian noises which varies from

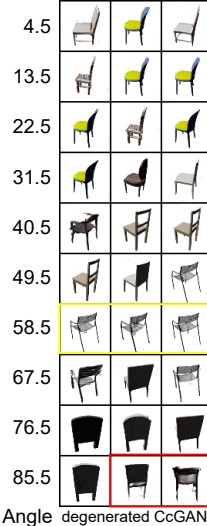

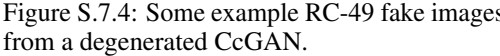

Figure S.7.4: Some example RC-49 fake images from a degenerated CcGAN.

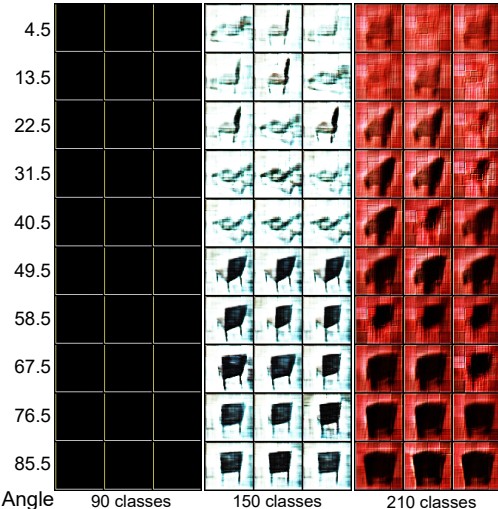

Figure S.7.5: Example RC-49 fake images from cGAN when we bin the yaw angle range into different number of classes.

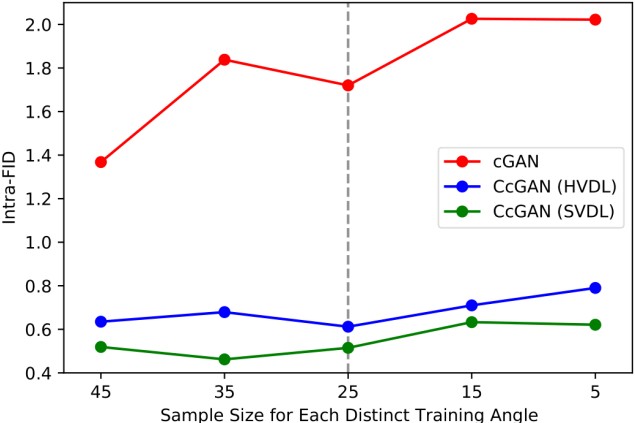

Figure S.7.6: Line graphs of Intra-FID versus the sample size for each distinct training angle. The grey vertical dashed line stands for the sample size used in the main study of the RC-49 experiment in Section 4.2. Two CcGAN methods substantially outperform cGAN no matter what the sample size for each distinct angle in the training set. The overall trend in this figure shows that a smaller sample size deteriorates the performance of both cGAN and CcGAN.

0 (no Gaussian noise) to 25 (the unit is degree). A large standard deviation corresponds to a weak correlation. The training setup is consistent with the main study in Section 4.2 except that we replicate the training set five times and randomly add Gaussian noises to the raw regression labels in the replicated training set. Therefore, each training sample has five noisy labels. We plot the line graphs of Intra-FID versus the standard deviation of Gaussian noise in Fig. S.7.7. From Fig. S.7.7, we can see that the performance of two CcGAN methods deteriorates as the standard deviation increases; however, the line graph of the performance of cGAN does not have a clear increasing or decreasing trend and the Intra-FID of cGAN always stays at a high level.

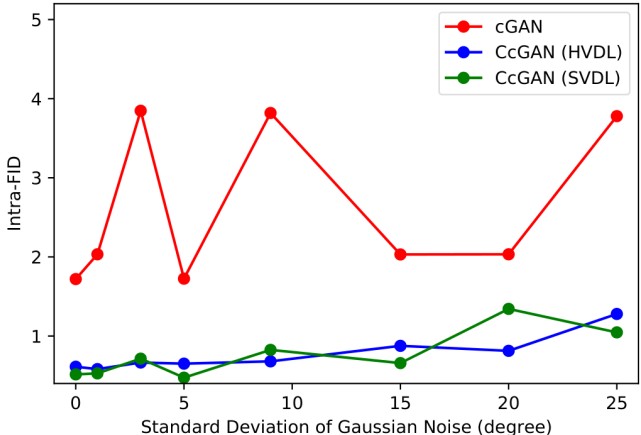

Figure S.7.7: Line graphs of Intra-FID versus the standard deviation of Gaussian noise. The overall trend in the figure shows that the performance of two CcGAN methods deteriorate as the standard deviation increases.

## S.8 MORE DETAILS OF THE EXPERIMENT ON THE UTKFACE DATASET IN SECTION 4.3

### S.8.1 DESCRIPTION OF THE UTKFACE DATASET

The UTKFace dataset is an age regression dataset (Zhang et al., 2017), with human face images collected in the wild. We use the preprocessed version (cropped and aligned), with ages spanning from 1 to 60. After data cleaning (i.e., removing images of very low quality or with clearly wrong labels), the overall number of images is 14760. Images are resized to $64 \times 64$. The histogram of UTKFace dataset w.r.t. ages 1-60 is shown in S.8.8.

From Fig. S.8.8, we can see UTKFace dataset is very imbalanced so the samples from the minority age groups are unlikely to be chosen at each iteration during the GAN training. Consequently, cGAN and CcGAN may not be well-trained at these minority age groups. To increase the chance of drawing these minority samples during training, we randomly replicate samples in the minority age groups to ensure that the sample size of each age is more than 200.

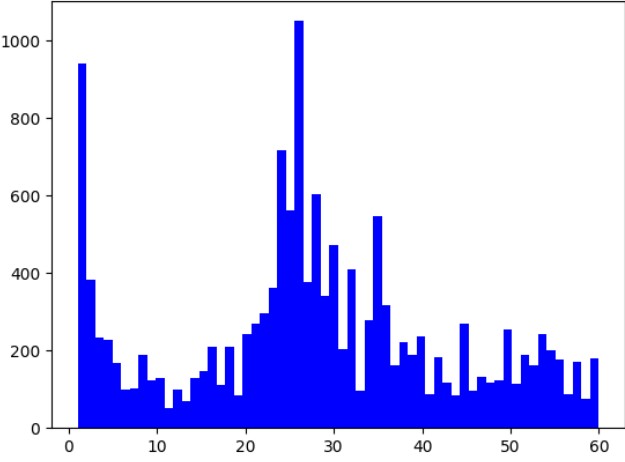

Figure S.8.8: The histogram of UTKFace dataset with ages range from 1 to 60.

### S.8.2 NETWORK ARCHITECTURES

The network architectures used in this experiment is similar to those in the RC-49 experiment. Please refer to our codes for more details about the network specifications.

### S.8.3 TRAINING SETUPS

The cGAN and CcGAN are trained for 40,000 iterations on the training set with the Adam (Kingma & Ba, 2015) optimizer (with $\beta_1 = 0.5$ and $\beta_2 = 0.999$), a constant learning rate $10^{-4}$ and batch size 512. The rule of thumb formulae in Section S.4 are used to select the hyper-parameters for HVDL and SVDL, where we let $m_\kappa = 1$.

### S.8.4 PERFORMANCE MEASURES

Similar to the RC-49 experiment, we evaluate the quality of fake images by Intra-FID, NIQE, Diversity, and Label Score. We also train an AE (bottleneck dimension is 512), a classification CNN, and a regression CNN on all images. Please note that, the UTKFace dataset consists of face images from 5 races based on which we train the classification CNN. The AE and both two CNNs are trained for 200 epochs with a batch size 256.

### S.8.5 EXTRA EXPERIMENTS

#### S.8.5.1 MORE LINE GRAPHS

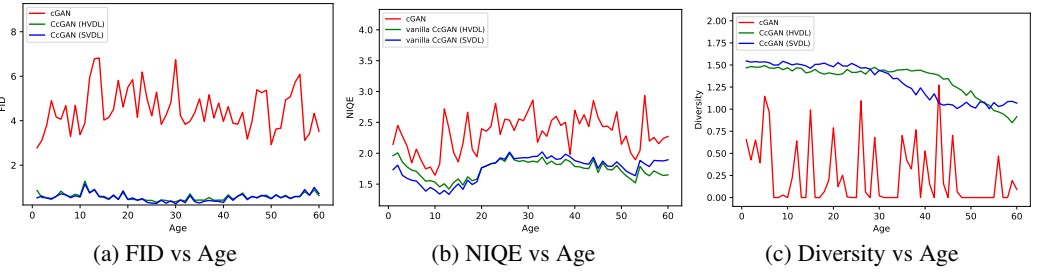

|   (a) FID vs Age   |   (b) NIQE vs Age   |   (c) Diversity vs Age   |

Figure S.8.9: Line graphs of FID/NIQE/Diversity versus ages on UTKFace. Figs. S.8.9(a) to S.8.9(c) show that two CcGANs consistently outperform cGAN across almost all ages. The graphs of CcGANs also appear smoother than those of cGAN because of HVDL and SVDL.

#### S.8.5.2 INTERPOLATION

To perform label interpolation experiments, we keep the noise vector $z$ fixed and vary label from age 3 to age 57 for CcGANs with HVDL and SVDL losses. The interpolation results are illustrated in S.8.10. As age $y$ increases, we observe the synthetic face gradually becomes older in appearance. This observation convincingly shows that both HVDL and SVDL based CcGANs do not simply memorize or overfit to the training set. Indeed, our CcGANs demonstrate continuous control over synthetic images with respect to ages.

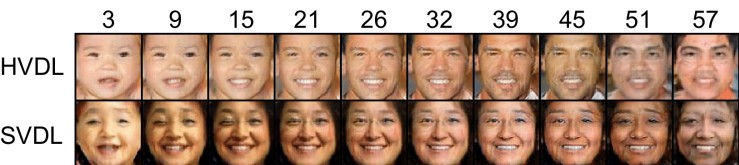

Figure S.8.10: Some examples of generated UTKFace images from CcGAN when the discriminator is trained with HVDL and SVDL. We fix the noise $z$ but vary the label $y$ from 3 to 57.

### S.8.5.3 DEGENERATED CCGAN

We consider the extreme cases of the proposed CcGANs on the UTKFace dataset. As shown in Fig. S.8.11, the degenerated CcGANs fails to generate facial images at some ages (e.g., 51 and 57) because of too small sample sizes.

### S.8.5.4 CGAN: DIFFERENT NUMBER OF CLASSES

In the last experiment, we bin samples into different number of classes based on ground-truth labels, in order to increase the number of training samples at each class. Then we train cGAN using samples from the binned classes. We experimented with two different bin setting, i.e., binning image samples into 60 classes and 40 classes, respectively. The results are reported in Fig.S.8.12. The results demonstrate cGANs consistently fail to generate diverse synthetic images with labels aligned with their conditional information. Moreover, the image quality is much worse than those from the proposed CcGANs. In conclusion, compared with existing cGANs, our proposed CcGANs have substantially better performance in terms of the image quality and diversity.

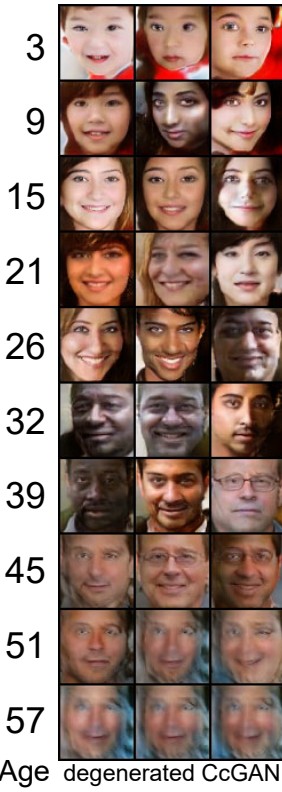

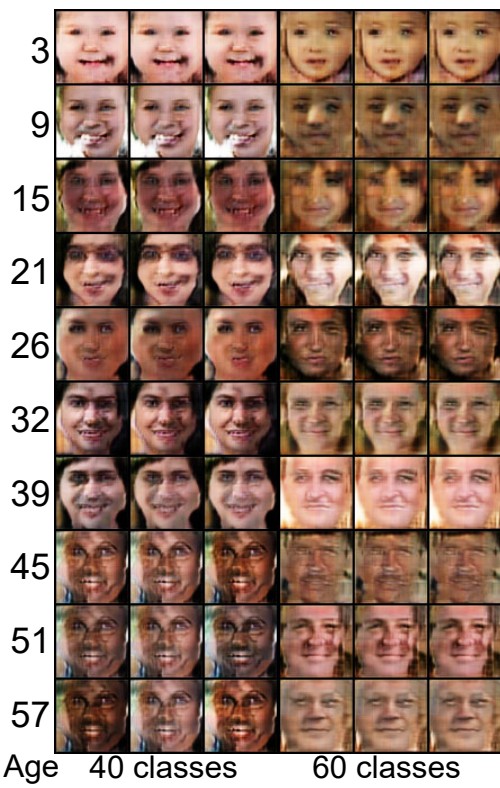

Figure S.8.11: Some example UTKFace fake images from a degenerated CcGAN.

Figure S.8.12: Example UTKFace fake images from cGAN when we bin the age range into different number of classes.

### S.8.5.5 TRAINING ON IMAGES FOR ODD AGES ONLY AND TESTING ON EVEN-NUMBERED AGES

In this section, we train cGAN and CcGAN on images for odd ages only and test them on even-numbered ages. Since the training set in this experiment is half of the one in the main study in Section 4.3, we reduce the number of iterations for the GAN training from 40,000 to 20,000 while other settings are unchanged. The quantitative results for cGAN and two CcGAN methods are summarized in Table S.8.3. From Table S.8.3, we can see two CcGAN methods are still much better than cGAN in terms of all metrics except Label Score, since CcGAN is designed to sacrifice some (not too much) label consistency for much better visual quality and diversity.

Table S.8.3: Training cGAN and CcGAN on images for odd ages only and testing them on even-numbered ages.

| Method | Intra-FID ↓ | NIQE ↓ | Diversity ↑ | Label Score ↓ |
|---|---|---|---|---|
| cGAN (30 classes) | $4.724 \pm 1.339$ | $2.763 \pm 0.384$ | $0.299 \pm 0.349$ | $\mathbf{9.114 \pm 7.398}$ |
| CcGAN (HVDL) | $\mathbf{0.724 \pm 0.161}$ | $\mathbf{1.795 \pm 0.230}$ | $\mathbf{1.133 \pm 0.257}$ | $10.341 \pm 3.931$ |
| CcGAN (SVDL) | $\mathbf{0.777 \pm 0.248}$ | $\mathbf{1.803 \pm 0.214}$ | $\mathbf{1.257 \pm 0.112}$ | $13.141 \pm 5.862$ |

### S.8.5.6 TRAINING WITH SMALLER SAMPLE SIZES

The histogram in Fig. S.8.8 shows that the UTKFace dataset is highly imbalanced. To balance the training data and also test the performance of cGAN and CcGAN under smaller sample sizes, we vary the maximum sample size for each distinct age in the training from 200 to 50. Note that, in the main study in Section 4.3, we do not restrict the maximum sample size. Since we have a much smaller sample size, we reduce the number of iterations for the GAN training from 40,000 to 20,000 and slightly increase $m_\kappa$ in Supp. S.4 from 1 to 2 (we therefore use a wider hard/soft vicinity). We visualize the line graphs of Intra-FID versus the maximum sample size for each age of cGAN and CcGAN in Fig. S.8.13. From the figure, we can clearly see that a smaller sample size worsens the performance of both cGAN and CcGAN. Moreover, the Intra-FID scores of cGAN always stay at a high level and are much larger than those of two CcGAN methods.

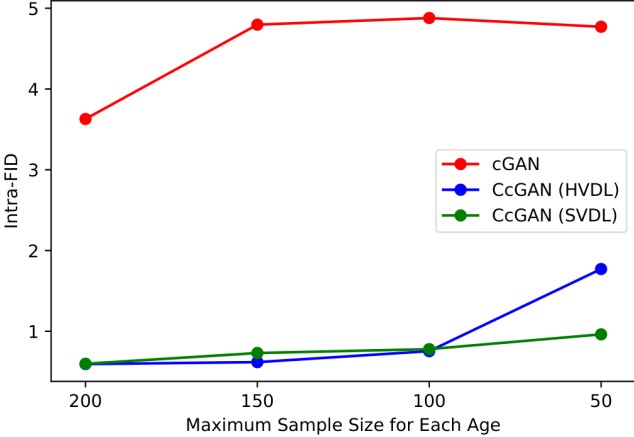

Figure S.8.13: Line graphs of Intra-FID versus the maximum sample size for each distinct angle in the training set.

### S.8.5.7 DIFFAUGMENT CANNOT SAVE CGAN IN THE CONTINUOUS SCENARIO

DiffAugment (Zhao et al., 2020) is a concurrent work which proposes differentiable transformations (i.e., translation, random brightness, contrast, saturation, and cutout) on both real and fake images during the GAN training. This method shows promising results in improving the performance of unconditional GANs (e.g., StyleGAN2 (Karras et al., 2020)) and class-conditional GANs (e.g., BigGAN (Brock et al., 2019)) when training samples are limited. However, DiffAugment is fundamentally different from CcGAN since it is designed for the unconditional and class-conditional scenarios rather than the continuous scenario. Even though incorporating DiffAugment into the cGAN training in the continuous scenario, the two problems **(P1)** and **(P2)** discussed in Section 1 are still unsolved. First, **(P1)** is still unsolved because DiffAugment does not provide a solution better than binning the regression labels into a series of disjoint intervals to tackle the problem that some regression labels do not exist in the training set. Second, since DiffAugment is designed for the unconditional and class-conditional scenarios where the number of distinct conditions is always finite and known, DiffAugment doesn't provide a solution to **(P2)**. Besides these two unsolved problems, another concern of DiffAugment in the continuous scenario is that the ordinal information in the regression labels is not utilized while our CcGAN implicitly uses this ordinal information to construct the soft/hard vicinity.

To support our arguments, we incorporate DiffAugment into the cGAN training in the UTKFace experiment while other settings are kept constant. When implementing DiffAugment, we use the official codes from the GitHub repository of DiffAugment [2]. The strongest transformation combination (Color + Translation + Cutout) is used in the cGAN training. Quantitative results from cGAN+DiffAugment are summarized in Table S.8.4 and some example images from cGAN+DiffAugment are shown in Fig. S.8.14. The quantitative results show that DiffAugment substantially improves the visual quality and diversity of the baseline cGAN; however, the performance of cGAN+DiffAugment is still much worse than that of the proposed two CcGAN methods. The visual results also support the quantitative evaluations. Therefore, cGAN+DiffAugment still does not solve the two fundamental problems in the continuous scenario, since it is not designed for this purpose.

Table S.8.4: Average quality of 60,000 fake UTKFace images from cGAN and CcGAN with standard deviations after the "$\pm$" symbol. "$\downarrow$" ("$\uparrow$") indicates lower (higher) values are preferred.

| Method | Intra-FID $\downarrow$ | NIQE $\downarrow$ | Diversity $\uparrow$ | Label Score $\downarrow$ |
|---|---|---|---|---|
| cGAN (60 classes) | $4.516 \pm 0.965$ | $2.315 \pm 0.306$ | $0.254 \pm 0.353$ | $11.087 \pm 8.119$ |
| cGAN (60 classes) + DiffAugment | $1.328 \pm 0.156$ | $2.077 \pm 0.245$ | $1.102 \pm 0.183$ | $11.212 \pm 8.329$ |
| CcGAN (HVDL) | $\mathbf{0.572 \pm 0.167}$ | $\mathbf{1.739 \pm 0.145}$ | $\mathbf{1.338 \pm 0.178}$ | $\mathbf{9.782 \pm 7.166}$ |
| CcGAN (SVDL) | $\mathbf{0.547 \pm 0.181}$ | $\mathbf{1.753 \pm 0.196}$ | $\mathbf{1.326 \pm 0.198}$ | $\mathbf{10.739 \pm 8.340}$ |

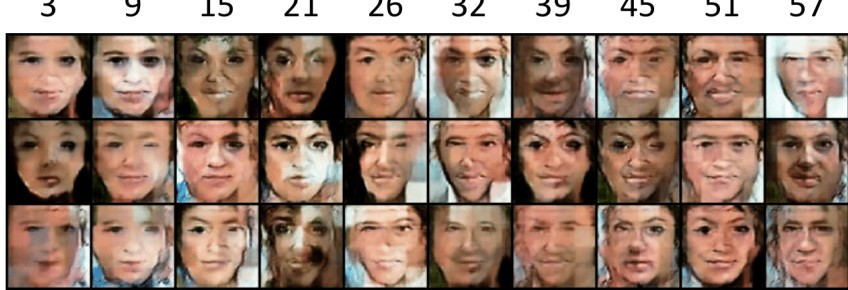

Figure S.8.14: Some example UTKFace fake images from cGAN+DiffAugment. Even with the help of DiffAugment, cGAN still has poor visual quality in the continuous scenario.

## S.9 POTENTIAL APPLICATIONS AND IMPACTS OF CcGANs

Generally, there are three label scenarios where we can apply CcGANs: Scenario I, mathematically continuous labels (e.g., angles); Scenario II, discrete but ordinal labels (e.g., ages); and Scenario III, discrete, categorical labels but sharing close relationships among different label categories (e.g., fine-grained bird image generation). CcGANs can have potential applications in all three scenarios. For example, in Scenario I, CcGANs could have potential impacts on autonomous driving which involves predicting the steering angle ( a continuous scalar) to have better controllability over autonomous cars. In Scenario II, the proposed methods are potentially meaningful in some medical applications. E.g., in medical experiments, an important task is cell counting, where the cell counting regression needs to predict the number of cells (i.e., ordinal integers) from a microscopic image. Even with limited microscopic cell images, the proposed CcGAN can generate visually synthetic and diverse microscopic images for the regression model training. In this way, CcGAN may help save tedious efforts of medical researchers in gathering microscopic images. In Scenario III, as suggested by AnonReviewer 5 (Q3), CcGAN could be used on some fine-grained image classification datasets, e.g., on the bird dataset where birds of different categories may share close similarities. The generated bird images can be used to enhance the fine-grained bird image classifiers, and potentially help us better recognize birds and protect the environment. More generally, CcGANs can be potentially used for image generation in regression datasets (associated with scalar labels y). In summary, CcGANs can cover a wide range of tasks and applications which could potentially benefit the society.

---

[2]https://github.com/mit-han-lab/data-efficient-gans

