# OpenReview forum: "CcGAN: Continuous Conditional Generative Adversarial Networks for Image Generation"
_ICLR.cc/2021/Conference — ICLR 2021 Poster_

### Official Review · AnonReviewer1 · 2020-10-27
**Interesting approach for an important topic, but ask for some clarifications**

**Rating:** 6
**Confidence:** 3

**Review:**

##### Summary
This paper tries to make the generative adversarial network handle continuous, scalar conditions. Specifically, in order to make it work, the author set an empirical estimate that every small perturbation to the condition y results in negligible changes to the conditional distribution.  The author shows comparative performance improvement over the basic cGAN on circular 2D Gaussians, RC-49, and UTKFace dataset.

##### Strength
First of all, this paper is well-written and easy to follow. This paper tackles one of the important topics in a generative network with appropriate assumptions and shows its relative effectiveness not only on a single dataset but also in various settings. It also provides rich details with the codes in supplementary.

##### Weakness
First of all, although the math and logic are quite easy to follow, it is a bit hard to guess, especially to the person who doesn't have a strong background in this domain, what exact behavior is enforced to the model. Also, along with the first issue, it is hard to predict this method's stability for some non-well curated/designed setting. Please check the question section for the details.

##### Question
1. As I mentioned in the weakness part, it is quite hard to imagine how this method will behave. Similar to Figure 1 of the original GAN paper (Goodfellow et al., in NeurIPS 2014), or Figure 1 of VEEGAN (Srivastava et al., in NeurIPS 2017), it would be great if the author can visualize how the distribution is matched and what kinds of behaviors are promoted/demoted by the loss.

2. As mentioned in (P1) of this paper, there are many real-world cases that there are zero or only one label exists. To show the effectiveness in such an imbalanced setting, I recommend presenting the result that (i) train people of odd ages only and tested with even-numbered ages and (ii) tested with fewer data in some of the ages (unlike the one in Figure S. VIII.5 or S.VIII.A).

3. Also, I also wonder what happened if the condition is not directly related (to see whether the label power is essential or not). To alleviate such curiosity, I recommend testing it by modifying the label on one of the experiments. For instance, you can try this method on the RC-49 generation, not with the angle but with the chair's volume.



##### Post-rebuttal

I thank the authors for their thorough comments and detailed explanations for each question. I carefully read the whole, and it helps me understand the entire decision and the processes. However, I would like to suggest two things, regardless of its acceptance, but to make their claim more attractive to general readers: (i) re-sort out the indexes (maybe after rebuttal but before submitting your camera-ready version), and (ii) update figures of the sample with a bit more realistic one (with training the model on larger batch size, for instance). I hope this review phase would make your paper more powerful. (disclaimer: I did not check the soundness of the mathematical equations thoroughly but did check all the rest)

---

> ### Author Response · Authors · 2020-11-22
> **Response to AnonReviewer1 (Part 1)**
>
> **Q1**: "As I mentioned in the weakness part, it is quite hard to imagine how this method will behave. Similar to Figure 1 of the original GAN paper (Goodfellow et al., in NeurIPS 2014), or Figure 1 of VEEGAN (Srivastava et al., in NeurIPS 2017), it would be great if the author can visualize how the distribution is matched and what kinds of behaviors are promoted/demoted by the loss."
>
> **A1**:  We appreciate the reviewer’s constructive suggestion, and we have added **Fig.1** (pp.4) in the revised version. To make the proposed ideas easier to understand, similar to the practice in two suggested references, we visualize the behaviors of the key steps of the two proposed losses in diagrams. To derive the HVDL and SVDL in Eqs. (9) and (10), we develop two estimates (i.e., HVE in Eq. (6) and SVE in Eq. (7)) of the joint distributions of $x$ and $y$. Firstly, we utilize KDEs to estimate the marginal distributions $p(y)$. We then estimate the key step, i.e., the estimate of conditional distributions $p(x|y)$ by utilizing samples in a hard/soft vicinity of y. We illustrate this step in **Fig. 1** (pp.4) in the revised version. Specifically, as illustrated in **Fig.1**, if we want to estimate $p(x|y)$ (the red curve) but we only have samples (red dots) drawn from $p(x|y_1)$ and $p(x|y_2)$ (the blue curves), then we do the estimation based on samples in a hard vicinity (defined by $y \pm \kappa$) or a soft vicinity (defined by the weight decay curve) of $y$. We hope the visualization/diagrams can better illustrate our ideas.
>
> ---
>
> **Q2**:  "As mentioned in (P1) of this paper, there are many real-world cases that there are zero or only one label exists. To show the effectiveness in such an imbalanced setting, I recommend presenting the result that (i) train people of odd ages only and tested with even-numbered ages and (ii) tested with fewer data in some of the ages (unlike the one in Figure S. VIII.5 or S.VIII.A)."
>
> **A2**:  We agree with the reviewer that there are zero or only one training image at some labels in real-world cases, and we appreciate the reviewer’s clever suggestions on additional experiments in (i) and (ii). In the revised manuscript, we have performed experiments and presented complete comparisons on the UTKFace dataset in **Supp. S.VIII.E.5** and  **Supp. S.VIII.E.6** (from pp.28-pp.29). We summarize our results and observations as follows.
>
> Experimental results on setting (i): We train cGAN and CcGAN on images for odd ages only and test them on even-numbered ages following the suggested setup (i). The quantitative results for cGAN and two CcGAN methods are summarized in **Table S.VIII.3** (pp.29). From **Table S.VIII.3**, we can see two CcGAN methods are still much better than cGAN in terms of all metrics except Label Score, since CcGAN is designed to sacrifice some (not too much) label consistency for much better visual quality and diversity.
>
> Experimental results on setting (ii): To balance the training data and also test the performance of cGAN and CcGAN under smaller sample sizes, we vary the maximum sample size for each distinct age in the training from 200 to 50. We visualize the line graphs of Intra-FID versus the maximum sample size for each age of cGAN and CcGAN in **Fig. S.VIII.13** (pp.29). From the figure, we can clearly see that a smaller sample size degrades the performance of both cGAN and CcGAN. Moreover, the Intra-FID scores of cGAN always stay at a high level and are much larger than those of two CcGAN methods.
> In summary, with additional suggested experimental setups (i.e., (i) and (ii)), we demonstrate the effectiveness and superiority of the proposed CcGANs over cGANs under different settings. We have also included the full experimental comparison/discussions in **Supp. S.VIII.E.5** and  **Supp. S.VIII.E.6** in the revised version.
>
> ---

---

> > ### Author Response · Authors · 2020-11-22
> > **Response to AnonReviewer1 (Part 2)**
> >
> > **Q3** "Also, I also wonder what happened if the condition is not directly related (to see whether the label power is essential or not). To alleviate such curiosity, I recommend testing it by modifying the label on one of the experiments. For instance, you can try this method on the RC-49 generation, not with the angle but with the chair's volume."
> >
> > **A3**:  We thank the reviewer for this interesting suggestion, and we certainly would like to explore more. Unfortunately, the RC-49 image dataset was generated by using 3D chair models from the ShapeNetCore repository which doesn’t have the volume annotation. Instead, we designed another experiment to evaluate the effect of label power on the performance of cGAN and CcGAN.
> >
> > To study how the strength of the correlation between the image $x$ and its label $y$ (i.e., the label power) influences the performance of cGAN and CcGAN, in this study, we randomly add Gaussians noises with a preset standard deviation to the raw regression labels in the training set. The strength of the correlation is controlled by the standard deviation of the Gaussian noises which varies from 0 (no Gaussian noise) to 25 (the unit is degree). A large standard deviation corresponds to a weak correlation. The training setup is consistent with the main study in **Section 4.2** except that we replicate the training set five times and randomly add Gaussian noises to the raw regression labels in the replicated training set. Therefore, each training sample has five noisy labels. We plot the line graphs of Intra-FID versus the standard deviation of Gaussian noise in **Fig. S.VII.7** (pp.26). From this figure, we can see that the performance of two CcGAN methods deteriorates as the standard deviation increases; however, the line graph of cGAN does not have a clear increasing or decreasing trend and the Intra-FID of cGAN always stays at a high level. This indicates the importance of label power in CcGANs which coincides with our intuition, since CcGANs exploits more on the label power (e.g., CcGANs implicitly utilize the ordinal information). In the revised version, we have included the experimental results and discussions in **Supp. S.VII.F.6** (from pp.24 to pp.26).
> >
> > ---

---

> ### Author Response · Authors · 2020-11-25
> **Response to AnonReviewer1 (Post-rebuttal)**
>
> We sincerely thank AnonReviewer1 for your constructive suggestions to further improve the readability/presentation of our work. We will update the paper in the next version following the suggestions (i) and (ii).

---

### Official Review · AnonReviewer2 · 2020-10-28
**Interesting idea, but experiments needs to be strengthen and polished.**

**Rating:** 5
**Confidence:** 3

**Review:**

 This work proposes to perform Conditional GAN with regression labels, thus to benefit the model with data-sufficient continuations generation (vs infinite distinct condition for generation).  Inspired by VRM, a CcGAN model is proposed to tackle the challenges in existing methods. (Eq.(6) ---> Eq.(10))  An error bound is also derived for the proposed new discriminator loss.

The idea to facilitate generation with continuous conditions is interesting and insightful. the experiments are also well-designed and analyzed to support the claim.  My questions are mainly in the following aspects:

1). Comparisons on more challenging data sets are suggested.  Also, as you argue the deficiency of requiring large sample and condition size in baseline models, comparison regarding the different number of, or scare training data are better to be presented.

2) Can you please explain your superiority and connection with [1].

[1]. Zhao, Shengyu, et al. "Differentiable augmentation for data-efficient gan training." arXiv preprint arXiv:2006.10738 (2020).

---

> ### Author Response · Authors · 2020-11-22
> **Response to AnonReviewer2 (Part 1)**
>
> "Interesting idea, but experiments needs to be strengthen and polished."
>
> With the experiments in the original submission and 6 extra experiments in the current version which are summarized in **General Response to Reviewers’ Comments**, we believe current experiments are already strong enough to demonstrate the effectiveness of the proposed CcGAN.
>
> ---
>
> **Q1**: " Comparisons on more challenging data sets are suggested. Also, as you argue the deficiency of requiring large sample and condition size in baseline models, comparison regarding the different number of, or scare training data are better to be presented."
>
> **A1**:  We agree with the reviewer that challenging datasets are very important in evaluating CcGAN performances. As a new research direction, indeed we have tried our best to find challenging datasets for our experiments. We firstly design the 2D-circular Gaussian dataset with analytical conditional distribution to perform experimental comparisons. In addition, we also experiment on two real-world image datasets: the UTKFace and RC-49 datasets. The UTKFace dataset requires us to do careful manual preprocessing since some face images have bad visual quality and watermarks.  We also generate the RC-49 dataset due to a lack of benchmark datasets in this research direction. As the first work in image generation conditional on continuous scalar labels, we focus on verifying the efficacy of the proposed methods and try to explain the story well. Indeed we plan to explore some other challenging real-world datasets/applications as our future work.
>
> As suggested by the reviewer, in the revision, we have additionally demonstrated the superiority of CcGANs by reducing the number of training samples at each scalar label. For instance, in the benchmark dataset RC-49, we vary the number of sample sizes for each distinct training angle gradually from 45 to 5. We include the experimental comparisons in **Fig. S.VII.6** (Intra-FID vs #sample sizes) in the updated version (in pp.25). As shown in **Fig. S.VII.6**, the proposed two CcGAN methods substantially outperform cGAN at different sample sizes for each distinct angle in the training set. Despite that both cGAN and CcGAN deteriorate with smaller sample sizes, we observe a sharper performance decrease trend on cGAN; while CcGAN only degrades its performance slightly. Even with a very small sample size (e.g., 5) at each distinct angle, CcGANs can still maintain good performances. By contrast, cGAN fails even with much more training samples (e.g., 45) at each angle. Besides, as suggested by AnonReviewer1 (Q2), we also have conducted experiments on the UTKFace dataset with reduced number of training samples. Experimental results show that CcGANs consistently outperform cGANs by a large margin on UTKFace under different setups. In summary, even with limited training data, the proposed CcGANs work well while cGANs could completely fail. Indeed, these additional results show that the proposed CcGANs outperform cGANs substantially in the scarce training data scenario. This is because, CcGANs are proposed to tackle the limited (or even 0) training sample problem encountered in the continuous conditional case. In the revision, we have included additional results in **Fig. S.VII.6** (pp.25), **Table S.VIII 3** (pp.29) and **Fig. S.VIII.13** (pp.29). Correspondingly, we have highlighted in red the detailed discussion regarding the experimental results/comparisons.
>
> ---

---

> > ### Author Response · Authors · 2020-11-22
> > **Response to AnonReviewer2 (Part 2)**
> >
> > **Q2**: "Can you please explain your superiority and connection with [1]"
> >
> > **A2**:   DiffAugment [1] is a concurrent work which proposes to do differentiable transformations (i.e., translation, random brightness, contrast, saturation, and cutout) on both real and fake images during the GAN training. This method shows promising results in improving the performance of unconditional GANs (e.g., StyleGAN2 [2]) and class-conditional GANs (e.g., BigGAN [3]) when training samples are limited. However, DiffAugment is fundamentally different from our proposed CcGAN since it is designed for the unconditional and class-conditional scenarios rather than the continuous scenario. Even though incorporating DiffAugment into the cGAN training in the continuous scenario, the two problems (**P1**) and (**P2**) discussed in **Section 1** are still unsolved. First, (**P1**) is still unsolved because DiffAugment does not provide a solution better than binning the regression labels into a series of disjoint intervals to tackle the problem that some regression labels do not exist in the training set. Second, since DiffAugment is designed for the unconditional and class-conditional scenarios where the number of distinct conditions is always finite and known, DiffAugment doesn't provide a solution to (**P2**). Besides these two unsolved problems, another concern of DiffAugment in the continuous scenario is that the ordinal information in the regression labels is not utilized while our CcGAN implicitly uses this ordinal information to construct the soft/hard vicinity.
> >
> > To support our arguments, in **Supp. S.VIII.E.7** (pp.29-pp.30), we incorporate DiffAugment into the cGAN training in the UTKFace experiment while other settings are kept the same. When implementing DiffAugment, we use the official codes from the GitHub repository of DiffAugment [4]. The strongest transformation combination (Color + Translation + Cutout) is used in the cGAN training. The quantitative result of cGAN+DiffAugment is summarized in **Table S.VIII.4** (pp.30) and some example images from cGAN+DiffAugment are shown in **Fig. S.VIII.14** (pp.30). The quantitative result shows that DiffAugment substantially improves the visual quality and diversity of the baseline cGAN; however, the performance of cGAN+DiffAugment is still much worse than that of the proposed two CcGAN methods. The visual results also support the quantitative evaluations. Therefore, cGAN+DiffAugment still doesn’t solve the 2 fundamental problems in the continuous scenario, since it is not designed for this purpose.
> >
> > ---
> >
> >
> > References:
> > [1]. Zhao, Shengyu, et al. "Differentiable augmentation for data-efficient gan training." arXiv preprint arXiv:2006.10738 (2020).
> > [2] Karras, Tero, et al. "Analyzing and improving the image quality of stylegan." Proceedings of the IEEE/CVF Conference on Computer Vision and Pattern Recognition. 2020.
> > [3] Brock, Andrew, Jeff Donahue, and Karen Simonyan. "Large scale gan training for high fidelity natural image synthesis." arXiv preprint arXiv:1809.11096 (2018).
> > [4] https://github.com/mit-han-lab/data-efficient-gans

---

### Official Review · AnonReviewer3 · 2020-10-30
**A novel continuous conditional GAN**

**Rating:** 7
**Confidence:** 4

**Review:**

*** Summary:

This paper proposes a novel continuous conditional GAN which takes continuous scalars (named regression labels) as conditions. There are two problems for continuous conditional GAN: (P1) cGANs with discrete labels are trained to minimize the empirical loss, but this fails for continuous conditions, because there might be few or even zero samples for many labels values. (P2) For continuous labels, the label cannot be embedded by one-hot encoding like discrete labels. The authors propose two solutions for the problems respectively: (S1) The authors give two types of estimate (HVE and SVE) of the joint probability used for calculating the loss, and further formulate new loss functions HVDL and SVDL, based on the estimated joint distribution. (S2) The authors propose new methods of encoding the label input: element-wise addition for the generator, and linear embedding layer for the discriminator. The authors further derive and prove the error bounds for a discriminator trained with HVDL and SVDL. Experiments are conducted on continuous label datasets, and the authors propose a new dataset with continuous labels. Comparisons are made with traditional cGANs with discrete labels.

*** Pros:

1. This paper is the first work on continuous conditional GAN. It solves the two key problems for generalizing traditional cGANs to continuous conditions, and empirically validates the effectiveness of the proposed approach.

2. The proposed approach is theoretically sound. The generalization from traditional cGANs with discrete labels to the novel continuous label conditional GANs is reasonable. The estimations (HVE and SVE) for deriving the new loss functions are clearly explained and mathematically elegant and solid.

3. Error bounds of the discriminator trained with HVDL and SVDL losses are derived and proved. This builds a strong and solid foundation of the proposed approach.

4. Experiments are conducted on various datasets. Reasonable results for generating synthetic data and real-world images are shown and compared with cGANs with discrete labels.

*** Cons:

1. The experiments are conducted on low-resolution images with SNGAN structure. In order to test the generalization ability of the proposed approach, would it be possible to test on other state-of-the-art structures such as BigGAN? And is it able to be trained to generate higher resolution images?

2. Conditional GANs with continuous label conditions is an interesting direction that has been rarely explored before. Since most real-world datasets are with discrete labels, and even for some labels that can be discrete, it might be impossible to label it in a continuous way, and we will assign discrete labels. Can you elaborate more on the potential application scenarios and potential impacts of the continuous cGANs, besides the viewpoint and age examples used in this paper?

*** Justification for rating

I think this paper works on an interesting direction of continuous conditional GANs, which has rarely been explored before. The proposed approach to generalize cGANs to continuous label conditions and the proposed solutions to the key problems are elegant and novel. The authors not only give clear explanations but also provides mathematical error bounds and proves. So I would recommend acceptance.

*** Questions for the rebuttal

Please address the questions in the Cons section.

*** After rebuttal

Thank the authors for the detailed explanation. After reading other reviewers' comments and the author feedback, I would like to keep my rating unchanged.

---

> ### Author Response · Authors · 2020-11-22
> **Response to AnonReviewer3**
>
> **Q1**: "The experiments are conducted on low-resolution images with SNGAN structure. In order to test the generalization ability of the proposed approach, would it be possible to test on other state-of-the-art structures such as BigGAN? And is it able to be trained to generate higher resolution images?"
>
> **A1**: We thank the reviewer for recognizing the novelty/contributions of our work, which we find very encouraging. Technically, the proposed algorithm can directly adopt more state-of-the-art network architectures to generate high-resolution images. This is because the method itself is a general framework, which is independent of image resolution or network architectures. In practice, generating high-resolution images (256x256 or 512x512) requires much more GPU memory than those for low-resolution images (64x64). Also, SOTA architectures adopt larger batch-size which necessitates more GPU memory (e.g., BigGAN uses batch-size as 2048). As our lab facility is limited, when we tried to experiment on BigGAN to generate high-resolution images, we found our lab GPU always prompted OOM error. We will open source all our codes/implementations, and we hope some researchers with adequate computational resources could help explore and experiment on generating high-resolution images.
>
> ---
>
> **Q2**: "Conditional GANs with continuous label conditions is an interesting direction that has been rarely explored before. Since most real-world datasets are with discrete labels, and even for some labels that can be discrete, it might be impossible to label it in a continuous way, and we will assign discrete labels. Can you elaborate more on the potential application scenarios and potential impacts of the continuous cGANs, besides the viewpoint and age examples used in this paper?"
>
> **A2**: We agree with the reviewer that many real-world datasets are with discrete labels and it is sometimes impossible to make them mathematically continuous (e.g., ages are always labeled as integers despite their continuous nature). In general, there are three label scenarios that we can apply CcGANs. Scenario I: mathematically continuous labels (e.g., angles); Scenarios II: discrete but ordinal labels (e.g., ages); and Scenario III: discrete, categorical labels but they share close relationships among different label categories (e.g., fine-grained bird image generation). CcGANs can have potential applications in three above scenarios. For instance, in Scenario I, CcGANs could have potential impacts on autonomous driving which involves predicting the steering angle ( a continuous scalar) to have better controllability over autonomous cars. In Scenario II, the proposed methods are potentially meaningful in some medical applications. E.g., in medical experiments, an important task is cell counting, where the cell counting regressor needs to predict the number of cells (i.e., ordinal integers) from a microscopic image. Even with limited microscopic cell images, the proposed CcGAN can generate visually synthetic and diverse microscopic images for the regressor training. In this way, CcGAN may help save many tedious efforts of medical doctors in gathering microscopic images. In Scenario III, as suggested by AnonReviewer 5 (Q3), CcGAN could be used on some fine-grained image classification datasets, e.g., on the bird dataset where birds of different categories may share close similarities. The generated bird images can be used to enhance the fine-grained bird image classifiers, and potentially help us better recognize birds and protect the environment. More generally, CcGANs can be potentially used for image generation on regression datasets (associated with scalar labels y), which can cover a wide range of tasks and applications. In the revision, we have elaborated and included more potential application scenarios and impacts of CcGANs in **Supp. S.IX** (pp.30).

---

### Official Review · AnonReviewer5 · 2020-11-04
**CcGAN: Continuous Conditional Generative Adversarial Networks for Image Generation**

**Rating:** 6
**Confidence:** 3

**Review:**

#########################################

Summary:

The paper focuses on conditional image generation with continuous label. Current methods utilize categorical conditions, which easily suffers form two problems: (1) the inferior performance when given few images for special category (2) failure of using continue label. In this paper, authors reformulate the class-conditional GANs, and  provide two new objectives (hard vicinal discriminator loss and soft vicinal discriminator loss ) and one generator loss. Both the discriminator loss and the generator loss is extended based on the vicinal risk minimization. Authors design interesting experiments to evaluate the proposed method.

#########################################

Pros:

+The paper explores new problem for conditional GANs. Specially the continues label never be studied, which is interesting for me.

+Authors leverage vicinal risk minimization to replace  current empirical generator loss, which is new viewpoint to investigate the conditional image generation.

+The first experiment is interesting, which is suitable to support the proposed method.

+The paper has good motivation, and is easy to follow.

#########################################

I have a few concerns which is as followiing:

1.  In page 2, authors mention '(1) our experiments in Section 4 show that this approach often makes cGANs collapse;' what is reason why current methods fail? Is it due to the imbalance of data for each category? or Is it normal issue of training GANs and cGANs?

2. I really like 'circular 2-d gaussians' experiment, which is smartly designed the proposed method. For the proposed method (E.q. 11 for generator), it has already  seen  the data which is corresponding to  test label (angle), since the real sample (Figure 1(a)) is localized in any angle and the used label (of the proposed method) for training is noised, which means the noised label is the test label.  Could authors try the real sample like one of Figure (b) which the real sample is not continues.

3.  I am wondering the proposed method could be used for fine-grand image generation. For example, the bird dataset[1] has 555 category,and some categories are close.

4. In paragraph: How is (P2) solved?. The new method to map the label seems similar to the label embedding. For example, the label embedding of BigGAN is also be updated. But here authors map the label by FC layers, which is corresponding to the learned label embedding. And the combination  of both the image embedding and the class embedding is similar to the one used in SNGAN and BigGAN.


[1] Grant Van Horn, Steve Branson, Ryan Farrell, Scott Haber, Jessie Barry, Panos Ipeirotis, Pietro Perona, and Serge Belongie. Building a bird recognition app and large scale dataset with citizen scientists: The fine print in fine-grained dataset collection. In CVPR, pages 595–604, 2015.

---

> ### Author Response · Authors · 2020-11-22
> **Response to AnonReviewer5 (Part 1)**
>
> **Q1**: "In page 2, authors mention '(1) our experiments in Section 4 show that this approach often makes cGANs collapse;' what is reason why current methods fail? Is it due to the imbalance of data for each category? or Is it normal issue of training GANs and cGANs?"
>
> **A1**: We appreciate the reviewer raising this interesting discussion on the failure of class-conditional GANs in the image generation conditional on continuous, scalar conditions (i.e., regression labels). Although more experiments on other datasets need to be done to confirm this observation, we feel it may be a normal issue in training class-conditional GANs in such continuous scenarios since we observe the same failure in both RC-49 and UTKFace experiments. We don’t think this issue is caused by the imbalance of data for each category since the RC-49 dataset is balanced but the class-conditional GAN still fails. Instead, we believe this failure occurs because there does not exist a good way to convert regression labels into class labels. In our experiments, to implement class-conditional GANs, regression labels are converted into class labels by binning regression labels into multiple disjoint intervals and each interval is taken as a class. Please note that, treating each distinct regression label as a class can be seen as a special case of the binning strategy when the width of the binned intervals equals zero.  We intuitively explain as follows how this ill-defined label conversion may cause the failure of class-conditional GANs.
>
> After binning regression labels into disjoint intervals, two consecutive intervals are separated by a cutoff point. If the primary assumption holds in this paper that a small perturbation to $y$ leads to a negligible change to $p(x|y)$, then samples at two sides of this cutoff point may come from similar conditional distributions (conditional on regression labels) but are grouped into two different classes. For example, if we bin the 60 distinct ages in UTKFace into three intervals, i.e., [1,20], [21, 40], [41, 60], the distribution of facial features at age 20 should be close to that at 21, while now images for age 20 and images for age 21 are in two different classes. Consequently, the image distribution conditional on these two classes may have a high overlap and this high overlap reduces the inter-class variation. On the other hand, images in each binned interval may have a high variation (i.e., intra-class variation) which may be comparable to or even larger than the inter-class variation. For example, in UTKFace, people of age 30 have different hair styles, different eye colors, different hair colors, different skin colors, different makeups, etc. It is very likely that such intra-class variation is much larger than the inter-class variation. Therefore, unlike the classes in common classification-oriented datasets (e.g., CIFAR-10, ImageNet, etc.), the binned intervals in UTKFace and RC-49 may have a large intra-class variation but a small inter-class variation, which makes it difficult for the class-conditional GANs to learn how the image distribution varies in terms of class labels. Therefore, class-conditional GANs may fail in the continuous scenario.

---

> > ### Author Response · Authors · 2020-11-22
> > **Response to AnonReviewer5 (Part 2)**
> >
> > **Q2**: "I really like 'circular 2-d gaussians' experiment, which is smartly designed the proposed method. For the proposed method (E.q. 11 for generator), it has already seen the data which is corresponding to test label (angle), since the real sample (Figure 1(a)) is localized in any angle and the used label (of the proposed method) for training is noised, which means the noised label is the test label. Could authors try the real sample like one of Figure (b) which the real sample is not continues."
> >
> > **A2**： We thank the reviewer’s interest in our ‘Circular 2-d gaussian’ experiment. Indeed, the CcGAN is to more effectively capture the continuous conditional distribution even with missing samples at certain labels. By sparsifying the number of Gaussians for the training data generation, we can simulate more non-continuous cases. For example, besides the number of Gaussians as 120 (dense enough to be continuous) in the original submission, we gradually decrease the number of Gaussians to make training examples more sparse and more non-continuous. Specifically, in the revision, we have used the number of Gaussians from 120 to 10 with step size 10, meanwhile keeping other settings in **Section 4.1** unchanged, to study the influence of the number of Gaussians used for training data generation on the performance of cGAN and CcGAN.
> >
> > As shown in **Fig. S.VI.1** (pp.22) in the revised version, we plot the line graphs of 2-Wasserstein Distance (in log scale) versus the number of Gaussians. From **Fig. S.VI.1**, we observe that **as the number of Gaussians decreases, the continuous scenario gradually degenerates to the categorical scenario, therefore the assumption that a small perturbation to $y$ results in a negligible change to $p(x|y)$ is no longer satisfied**. Consequently, the 2-Wasserstein distances of the proposed two CcGAN methods gradually increase and eventually surpass the 2-Wasserstein distance of cGAN when the number of Gaussians is small (e.g., less than 40). Note that reducing the number of Gaussians in the training data generation won't improve the performance of cGAN in the testing because a lot of angles seen in the testing stage (we evaluate each method on 360 angles) do not appear in the training set. We have included the additional experimental comparison and discussions in **Supp. S.VI.D.1** (pp.21-pp.22) in our revised manuscript.
> >
> > ---
> >
> > **Q3**: "I am wondering the proposed method could be used for fine-grand image generation. For example, the bird dataset[1] has 555 category,and some categories are close."
> >
> > **A3**: We thank the reviewer for raising this interesting research direction. We believe the proposed ideas can be generalized back to some class-conditional image generation scenarios and used for fine-grained image generation, e.g., on the bird dataset [1] where birds of different categories may share close similarities. Modifications and different implementations will be needed. Our intuitive idea is as follows: When generating images conditional on a class $y$, we first identify samples in a hard/soft vicinity of $y$ and then use these samples to estimate the image distribution conditional on the class $y$. Please note that the samples in the vicinity of $y$ may come from other classes. To construct the hard/soft vicinity of $y$, we need to evaluate the similarity between a sample in the dataset and the class $y$. Therefore, the difficulty here is how to define such similarity and how to compute it efficiently in practice (especially when we have a large sample size). Due to the time and page limit, we plan to explore more in this direction in the future.
> >
> > Reference: [1] Grant Van Horn, Steve Branson, Ryan Farrell, Scott Haber, Jessie Barry, Panos Ipeirotis, Pietro Perona, and Serge Belongie. Building a bird recognition app and large scale dataset with citizen scientists: The fine print in fine-grained dataset collection. In CVPR, pages 595–604, 2015.

---

> > > ### Author Response · Authors · 2020-11-22
> > > **Response to AnonReviewer5 (Part 3)**
> > >
> > > **Q4**: "In paragraph: How is (P2) solved?. The new method to map the label seems similar to the label embedding. For example, the label embedding of BigGAN is also be updated. But here authors map the label by FC layers, which is corresponding to the learned label embedding. And the combination of both the image embedding and the class embedding is similar to the one used in SNGAN and BigGAN."
> > >
> > > **A4**: To solve (**P2**), we propose a novel label input method that is suitable for the continuous conditional embedding scenario. Specifically, for the generator, we add the scalar label $y$ element-wisely to the output of its first linear layer. For the discriminator D, we firstly train an extra linear layer together with D to embed $y$ in a latent space. We then incorporate the embedded label into D using label projections. We would like to clarify that the proposed label embedding method is different from that used in SNGAN or BigGAN. In SNGAN and BigGAN, the label embedding method was designed for the class-conditional scenario, where the number of distinct regression labels is known and fixed (e.g., 1000 for ImageNet class) in advance. The class labels firstly need to be one-hot encoded prior to performing the learned label embedding. However, in the continuous conditional case, the number of distinct labels is not known (since distinct labels are infinite), which makes the existing class-conditional label embedding methods infeasible in our continuous cases. By contrast, the proposed label embedding method does not need this information and it is therefore more applicable to handle the continuous conditional label embedding situation. Interested readers may refer to our Supplementary Material **Supp.S.III** and our codes for more details on our label embedding method.

---

### Author Response · Authors · 2020-11-22
**General Response to Reviewers’ Comments**

We sincerely thank all reviewers for their valuable comments and constructive suggestions. As suggested, we have revised the paper, adding more experimental results on suggested experimental setups (e.g., varying sizes of training data, evaluation on the label power), comparison with a concurrent literature DiffAugment [1], visualization of the proposed methods with diagrams for better presentation, and elaboration on some potential applications and impacts of CcGANs. Experimental results demonstrate that the proposed CcGANs consistently outperform cGANs by a large margin in the continuous scenario. In the revision, we have highlighted important changes in red, and we summarize our revisions as follows:

(1) We visualize the key ideas behind HVE/SVE with illustrative diagrams (in **Fig. 1** in pp.4). A key step to develop HVE/SVE is to utilize samples in the hard/soft vicinity of $y$ to estimate $p(x|y)$. As suggested by AnonReviewer1 (Q1), for better presentation, we visualize this key step with diagrams (see **Fig. 1**) in our revised version.

(2) We have added more experiments with varying numbers of Gaussians (i.e., from 120 to 10) for training in the Circular 2-D Gaussian experiment, following AnonReviewer5’s suggestion (Q2). Please find the performance comparisons and result discussion of  this setting in **Supp. S.VI.D.1** (pp.21-pp.22).

(3) We have added experiments on RC-49 by varying the sample size for each distinct angle in the training set from 45 to 5 to make the dataset more challenging. This experiment setup follows the suggestion of AnonReviwer2 (Q1). Please find the experimental results in **Supp. S.VII.F.5** (pp.24).

(4) We have conducted extra experiments on RC-49 by perturbing ground-truth labels to evaluate the label power, as suggested by AnonReviwer2 (Q2). Experimental results have been presented in **Supp. S.VII.F.6** (pp.24-pp.26).

(5) We have conducted extra experiments on UTKFace with training images at odd ages and evaluate/generate images at even-numbered ages, following the suggestion of AnonReviewer1 (Q2). We have reported the comparison results in **Supp. S.VIII.E.5** (pp.28-pp.29).

(6) We have added more experiments on UTKFace by varying the sample size at each age in the training set from 200 to 50. This experimental setting serves to alleviate the concern of AnonReviewer1 (Q2) and AnonReviewer2 (Q1). We have included the experimental results in **Supp. S.VIII.E.6** (pp.29).

(7) We have further evaluated cGAN by incorporating a concurrent work DiffAugment [1] (a SOTA technique to train unconditional or class-conditional GANs when training samples are limited) into the cGAN training, as suggested by AnonReviewer2 (Q2). Experimental results show that DiffAugment cannot save cGAN in this continuous scenario. We have presented the experimental results and clarified the fundamental distinctions between DiffAugment [1] and the proposed CcGANs in **Supp. S.VIII.E.7** (pp.29-pp.30).

(8) Following the AnonReviewer3’s suggestions, we have elaborated and included more potential application scenarios and impacts of CcGANs in **Supp. S.IX** (pp.30).

References
[1]. Zhao, Shengyu, et al. "Differentiable augmentation for data-efficient gan training." arXiv preprint arXiv:2006.10738 (2020)

---

### Author Response · Authors · 2021-01-19
**Camera-ready version uploaded**

We have uploaded the camera-ready version in which we (1) add the acknowledgments; (2) change the style of the section indices in the appendix from Roman to Arabic; (3) fix a few typoes.

Thank all reviewers and program chairs for their valuable comments and constructive suggestions.

---

### Decision · Program_Chairs · 2021-01-07
**Final Decision**

**Decision:**

Accept (Poster)

**Comment:**

The submission proposes a novel conditional GAN formulation where continuous scalars (named regression labels) are fed into the GAN as a conditioning variable. Since cGANs with discrete labels are trained to minimize the empirical loss, they fail for continuous conditions, because there might be few or even zero samples for many labels values and also the label cannot be embedded by one-hot encoding like discrete labels. As a solution, the authors propose new methods of encoding the label.

The paper received a clear accept, two weak accepts and a weak reject. As agreed by all the reviewers, the paper proposes an interesting framework to eliminate some weaknesses of GANs. The rebuttal adequately addresses the reviewer comments and hence the meta reviewer recommends acceptance.